# Vertical Validation: Evaluating Implicit Generative Models for Graphs on Thin Support Regions

**Mai Elkady**[1]  **Thu Bui**[1]  **Bruno Ribeiro**[1]  **David I.Inouye**[2]

[1]Computer Science Dept., Purdue University, West Lafayette, Indiana, USA
[2]Electrical and Computer Engineering Dept., Purdue University, West Lafayette, Indiana, USA

## Abstract

There has been a growing excitement that implicit graph generative models could be used to design or discover new molecules for medicine or material design. Because these molecules have not been discovered, they naturally lie in unexplored or scarcely supported regions of the distribution of known molecules. However, prior evaluation methods for implicit graph generative models have focused on validating statistics computed from the thick support (e.g., mean and variance of a graph property). Therefore, there is a mismatch between the goal of generating novel graphs and the evaluation methods. To address this evaluation gap, we design a novel evaluation method called Vertical Validation (VV) that systematically creates thin support regions during the train-test splitting procedure and then reweights generated samples so that they can be compared to the held-out test data. This procedure can be seen as a generalization of the standard train-test procedure except that the splits are dependent on sample features. We demonstrate that our method can be used to perform model selection if performance on thin support regions is the desired goal. As a side benefit, we also show that our approach can better detect overfitting as exemplified by memorization.

## 1 INTRODUCTION

Over the past decade, significant progress has been achieved in enhancing implicit generative models (GANs [Goodfellow et al., 2014], VAEs [Kingma and Welling, 2022], and diffusion models [Sohl-Dickstein et al., 2015, Ho et al., 2020]), leading to their extensive use in diverse domains like image and graph generation. In the image generation domain, efforts have been made to standardize evaluation metrics [Wang et al., 2004, Zhang et al., 2018, Heusel et al., 2017] for comparing the effectiveness of different implicit generative models. However, the graph generation domain has yet to adopt a similar standardization. Moreover, while visual inspection of an image can reveal much about its semantic characteristics, this cannot be applied to graphs. Perhaps, more importantly, the application of graph generative models in different areas is quite different than image generators. Instead of aiming to generate an image that looks like others, most graph generative models are designed in hopes that they will be able to generate novel yet interesting graphs, e.g., new molecules with specific properties.

While extrapolating far from the known distribution of graphs is indeed challenging, there's potential for generative models to explore novel graphs within underexplored regions of the graph space by leveraging patterns observed in existing graphs. We illustrate this concept and our proposed evaluation methodology in Figure 1, using molecules as an example. Thick support regions represent known molecules, while thin support regions denote the space of novel graphs. We note that unlike this toy 2D illustration, real graph distributions (like image distributions) are expected to have many areas of thin support in high dimensions though they may be difficult to identify or characterize. Thus, the question arises: *How can we measure a graph generative model's ability to generate novel graphs on thin support regions?*

The most intuitive and potentially ideal evaluation approach would involve computing the negative log-likelihood on a test dataset. This metric, relying on the KL divergence, is inherently sensitive to thin support regions. However, for modern implicit generative models, log-likelihood is difficult to compute exactly or even approximate well.

Given these challenges, most recent evaluations of generative models seek to compare statistics between generated samples and a held-out test set. A simple approach is to merely compare the means of these distributions or the means of various graph properties. Extending the difference in means to the worst case difference between the expecta-

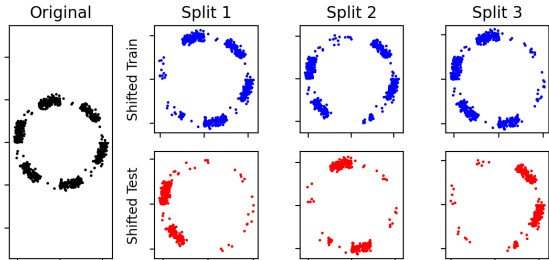

Figure 1: VV systematically thins the distribution in a certain region for training (top row) and then evaluates whether the generated samples in the thinned region after reweighting matches the complementary held-out test dataset (bottom row). In contrast, standard evaluations will seek to match the macro properties (e.g., mean) of this distribution which emphasizes the regions of thick support. The original data (left) illustrates both thick support regions (i.e., areas with many samples) and thin support regions (i.e., areas with very few samples).

tion of a function is known as Maximum Mean Discrepancy (MMD). The current and most commonly used standard procedure for evaluating graph generative models is to compute the MMD for the degree, clustering coefficient and orbit count distributions between the generated samples and a held-out set [Niu et al., 2020, Chen et al., 2021, Liao et al., 2019a, Hoogeboom et al., 2022, Vignac et al., 2022]. However, these mean-based approaches focus on the regions of thick support where the most mass is. Thus, they can fail to detect a generative models' performance on the thin support regions—the exact regions where novel graphs could exist. We illustrate this problem in more detail Appendix A.

To address this evaluation gap, we focus on matching the statistics (e.g., KS [Kolmogorov-Smirnov et al., 1933]) of systematically constructed thin regions of support as illustrated in Figure 1. Inspired by the classic train-test split idea, we develop a novel method to "vertically" split the graph dataset into train and test datasets depending on one graph property. Then, after training, we reweight generated samples and compare them to the corresponding held-out test dataset. At a high level, our evaluation approach, called Vertical Validation (VV), artificially simulates a thin region, but then has ground truth samples from this thinned region to compare against. After reweighting, any metric that can handle weights could be used to compare the generated samples to the held-out samples. We choose the average KS statistic along graph property distributions though other metrics could be used within our framework. This procedure enables the evaluation of the generation capabilities in localized thin support regions rather than focusing on the thick support regions. We summarize our contributions as follows:

1. We develop a novel "vertical" train-test splitting ap-

proach that systematically creates thin support in the training data while the testing data has thick support in this region. This can be applied to arbitrary 1D distributions and includes two hyperparameters that control the split sharpness and thickness of full support.

2. We combine this split procedure with a reweighting step to form a novel methodology for evaluating the ability to generate data in thin support regions. We prove that this metric instantiated with the KS statistic is consistent.

3. We empirically validate our VV approach for model selection in the thin support regime of synthetic datasets and then apply VV to compare representative graph generative models on two popular graph datasets.

## 2 BACKGROUND AND RELATED WORK

**Evaluating Graph Generative Models** Several methods have been used for evaluating the performance of graph generative models. Some methods can be used for all graph types. These methods include novelty, uniqueness, Wasseretian distance between generated samples and a held-out set, Maximum Mean Dependency (MMD) for the degree, clustering coefficient and orbit count distributions between the generated samples and a held-out set as used by Liao et al. [2019b], Martinkus et al. [2022], Vignac et al. [2022], Hoogeboom et al. [2022].

On the other hand, some of the metrics are specific for molecular generation tasks, such as the Frechet ChemNet Distance(FCD) introduced by Preuer et al. [2018], or the Neighbourhood subgraph pairwise distance kernel (NSPDK) MMD introduced by Costa and Grave [2010]. Other metrics include the percentage of atom stability, molecule stability, validity of generated molecules as used by Hoogeboom et al. [2022], Vignac et al. [2022] and others.

As one critique of prior evaluations, O'Bray et al. [2022] noticed that metrics based on MMD were sensitive to the choice of the kernel functions, the parameters of kernel, and the parameters of the descriptor function. Thompson et al. [2022] also noted that current evaluation methods do not accurately capture the diversity of the generated samples, which lead them to propose their own approach based on using the graph embedding produced by GIN [Xu et al., 2018] and calculating metrics on that embedding to better capture diversity.

Southern et al. [2023] recently proposed the use of curvature descriptors and topological data analysis for a more robust and expressive metric for evaluating graph generative models but does not specifically consider thin support regions. Despite this progress, there are still deficiencies in the current metrics particularly, when it comes to measuring the ability of the model to generate data in thin support regions.

**Related Train-Test Validation Methods** While classic cross validation methods sample form i.i.d. splits [Arlot and Celisse, 2009], our approach creates splits that are nearly out-of-distribution, which means that there is a distribution shift between train and test. Evaluating models under distribution shift has been studied for supervised learning under the names of domain adaptation (DA) [Farahani et al., 2020] and domain generalization (DG) [Koh et al., 2020]. In both cases, the accuracy metric is evaluated on a test distribution that is different from the training distribution. However, both DA and DG primarily consider supervised learning tasks while we consider generative models. Thus, our approach can be viewed as a type of distribution shift evaluation for generative models.

In a similar vein, Bazhenov et al. [2023] propose a method for splitting the *nodes* of a graph into in-distribution and out-of-distribution nodes based on structural properties. This splitting enables the evaluation of node-level prediction tasks under distribution shifts. We differ from this work because we split along graphs instead of along nodes, and ours is aimed at evaluating generative models while theirs is focused on node-level tasks.

# 3 MEASURING GENERALIZATION FOR IMPLICIT GRAPH GENERATIVE MODELS

Our proposed vertical validation method (VV) can be viewed as a generalized version of cross validation with two main steps. First, we propose a new way to generate biased train-test splits [1] that are dependent on a chosen graph property (e.g., average node degree), which we call the *split property*. In particular, our train-test splits thin some regions of the support along the split property. Second, we propose a meta evaluation metric that reweights the generated samples to unbias them and then compares them to the held-out samples using a two-sample metric on other graph properties. This second step is needed because the training sample distribution is different than the held-out test distribution based on our biased train-test splitting. The biased split and reweighting is carefully controlled using only the 1D distributions of the split property.

Under these circumstances, if the model can still produce "good" samples in a region that had thin support, we know that the model can generalize well. Conversely, if the model does not capture the underlying smoothness of the true distribution, it may struggle to generate realistic samples in regions with reduced training data support (underfitting) or it will memorize such data (overfitting).

---

[1] We use the term "split" here instead of "fold" because "fold" may seem like uniform splitting.

**Notation** To describe our method, we begin by introducing some notations that we will use for the rest of the paper. Let $p(G)$ denote the true graph distribution of graphs and let $G \sim p(G)$ be a random variable representing a graph, which includes the graph itself and any node or edge attributes if available. Let $m$ be the number user-specified graph properties of interest (e.g., average node degree). These properties are defined by deterministic functions of the graph denoted by $h_\ell : \mathbb{G} \to \mathbb{R}$, where $\mathbb{G}$ is the space of all valid graphs and $\ell \in \{1, \cdots, m\}$.

Let $\mathbf{h}(G) : \mathbb{G} \to \mathbb{R}^m$ be the vector function that maps $G$ to its $m$ graph property values, i.e., $\mathbf{h}(G) = (h_1(G), h_2(G), \cdots, h_m(G))$. Furthermore, let $\mathbf{Z} = \mathbf{h}(G)$, where the distribution of $Z$ is the pushforward of the graph distribution under $\mathbf{h}$. Let the true marginal CDFs of each dimension of $\mathbf{Z}$ be denoted by $F_{Z_\ell}(Z_\ell)$. We now define the random vector $\mathbf{U} \in [0,1]^m$, where each element is the corresponding CDF value of $Z_\ell$, i.e., $U_\ell = F_{Z_\ell}(Z_\ell) = F_{h_\ell(G)}(h_\ell(G)), \forall \ell \in \{1, 2, \cdots, m\}$.

**Train-Test Split Notation** Let $(G_i)_{i=1}^n$ be our given dataset which are i.i.d. samples from $p(G)$ and where each $G_i$ is a random variable. For $k$-fold cross validation, we introduce $m$ split variables corresponding to each graph property, denoted as $S_{i,1}, S_{i,2}, \cdots, S_{i,m} \in \{1, 2, \cdots, k\}$, that indicate the test split for the $i$-th graph using the $\ell$-th split property. Let $S_{i,\ell} \sim p(S|G_i, h_\ell)$, where $p(S|G_i, h_\ell)$ denotes the splitting distribution which can depend on the graph $G_i$ and the $\ell$-th graph property. Moving forward, $i$ will be used for the index of the graphs in the sequence $(G_i)_{i=1}^n$, i.e. $i \in \{1, 2, \cdots n\}$, and $j$ is used for the index of the splits, i.e. $j \in \{1, \cdots, k\}$. Given these split variables, the held-out dataset from $(G_i)_{i=1}^n$ of the $j$-th split and $\ell$-th split property will be denoted as $\mathcal{G}_{\text{held}}^{(\ell,j)} = \{\!\{G_i | \forall i, S_{i,\ell} = j\}\!\}$, where double curly braces $\{\!\{\}\!\}$ denotes a multi-set indicating that any element can have a multiplicity more than 1. The corresponding training dataset will be denoted $\mathcal{G}_{\text{train}}^{(\ell,j)} = \{\!\{G_i | \forall i, S_{i,\ell} \neq j\}\!\}$. Finally, let $\{\bar{G}_i^{(\ell,j)}\}_{i=1}^{n_{\ell,j}}$ be $n_{\ell,j}$ i.i.d. samples from $q(\bar{G}^{(\ell,j)} | \theta = \theta^*_{\Omega(\mathcal{G}_{\text{train}}^{(\ell,j)})})$, which denotes the generated graph distribution using a training algorithm $\Omega$ that only has access to $\mathcal{G}_{\text{train}}^{(\ell,j)}$ and let the generated dataset be denoted by $\mathcal{G}_{\text{gen}}^{(\ell,j)} = \{\!\{\bar{G}_i^{(\ell,j)} | \forall i\}\!\}$.

## 3.1 STEP 1: SHIFTED SPLITTING

In the first part of our framework, we need to define the distribution $p(S|G_i, h_\ell)$ to create $k$ biased splits for a given graph $G_i$ and $\ell$-th graph property. By biased splits we mean that the split variable depends on the graph, i.e., $P(S_{i,\ell}|G_i) \neq P(S_{i,\ell})$. For simplicity, we will assume that the conditional distribution is only dependent on the value of the $\ell$-th graph property, i.e., $p(S|G_i, h_\ell) = p(S|Z_{i,\ell})$. This will enable us to focus on a 1D distribution for splitting and reweighting. Many distributions of $P(S_{i,\ell}|G_i)$ could

give biased splits, but we wanted both a generic and balanced splitting method, and hence we incorporate the two constraints mentioned below.

*For the first constraint*, we want our method to work generically for any arbitrary distribution of $Z_{i,\ell}$. Thus, instead of using $Z_{i,\ell}$ directly, we only consider the CDF value of $Z_{i,\ell}$, i.e., we assume $p(S_{i,\ell}|G_i, h_\ell) = p(S_{i,\ell}|U_{i,\ell})$, where $U_{i,\ell} = F_{Z_{i,\ell}}(Z_{i,\ell}) = F_{h_\ell(G_i)}(h_\ell(G_i))$. This means that the splitting only depends on the rank of the $\ell$-th graph property rather than a specific value and thus it can be generically applied to any graph property. Essentially this constraint acts to restrict the space of distributions to those that only depend on $U_{i,\ell}$ making it applicable to any property distribution.

*For the second constraint*, we want the splits to have equal sizes in expectation to ensure that the splits are balanced. To achieve such effect, the marginals of $S$ must be uniform, i.e., we must ensure that $p(S_{i,\ell}) = 1/k$. To satisfy this last constraint, we notice that we can decompose the conditional distribution via Bayes rule $p(S_{i,\ell}|U_{i,\ell}) = \frac{p(S_{i,\ell})p(U_{i,\ell}|S_{i,\ell})}{p(U_{i,\ell})}$, where $p(S_{i,\ell}) = 1/k$ to ensure equal splits and $p(U_{i,\ell})$ is the uniform distribution by the fact that $U_{i,\ell}$ is based on the CDF of $Z_{i,\ell}$. Thus, we can choose any distribution for $p(U_{i,\ell}|S_{i,\ell})$ that satisfies the constraint that the marginal is uniform, i.e., $\sum_j p(S_{i,\ell} = j)p(U_{i,\ell}|S_{i,\ell})$ is uniform. In other words this constraint can be simplified to finding a component distribution whose mixture is a uniform distribution and whose weights are equal to $p(S_{i,\ell})$. One such choice of $p(U_{i,\ell}|S_{i,\ell})$ could be disjoint uniform splits corresponding to the quantiles of $U_{i,\ell}$, i.e., $p(U_{i,\ell}|S_{i,\ell} = j) = p_{\text{Unif}[\frac{j-1}{k}, \frac{j}{k}]}(U_{i,\ell})$, where $p_{\text{Unif}[a,b]}(U) = \frac{1}{b-a}$ denotes a Uniform distribution between the interval $[a, b]$.

However, there are two issues with quantile splits: (1) sharp cutoff for splits would create unnatural sharp edges in the training and test distributions, and (2) this would mean zero support on parts of the distribution, which would mean generative models would have to extrapolate beyond their training data—something that cannot be easily done with current methods (see Figure 3d for an illustration for quantile splits). To address the issues with the uniform quantile splits, we can resolve to using a different distribution for $p(U_{i,\ell}|S_{i,\ell})$ that satisfies the constraint mentioned above.

**Using Beta Distributions to Create Smoothed Quantile Splits:** For the first issue of unnaturally sharp distribution edges, inspired by empirical Beta copula models [Segers et al., 2017], we first note that a simple mixture of Beta distributions will have a uniform marginal distribution—the exact property we need for splitting (ie. satisfies $U_{i,\ell} \sim \text{Uniform}[0, 1]$ ). Specifically, a mixture of $k$ Beta distributions with parameters defined as: $\alpha_j = j$ and $\beta_j = k + 1 - j$ and for $j \in \{1, 2, \cdots, k\}$ will have a uniform distribution [Segers et al., 2017], i.e., $\sum_j p_{\text{Beta}[\alpha_j,\beta_j]}(U) = p_{\text{Unif}[0,1]}(U)$. Thus we

can choose $p(U_{i,\ell}|S_{i,\ell}) = p_{\text{Beta}[\alpha_j,\beta_j]}(U_{i,\ell})$, which will lead to $\sum_j p(S_{i,\ell} = j)p_{\text{Beta}[\alpha_j,\beta_j]}(U_{i,\ell}|S_{i,\ell} = j) = p_{\text{Unif}[0,1]}(U_{i,\ell}) = p(U_{i,\ell})$.

However, we still notice some potential issues with this approach, first the intervals have more overlap than what we want, which will be the case if $k$ is relatively small, and second this approach doesn't guarantee a support for all the regions in the current split. To deal with the first concern, we propose adding a sharpness scale ($\psi$) that acts to sharpen the edges of the distribution (i.e., making the splits more vertical). The sharpness scale identifies the number of adjacent Beta distributions to mix together for a single split distribution. Specifically, if we use $\psi \cdot k$ to be the total number of Beta distributions, then we can let each of the split distributions be mini mixtures of adjacent distributions, i.e., $p_{\text{BetaMix}}(U_{i,\ell}|S_{i,\ell} = j) = \frac{1}{\psi} \sum_{a=1}^{\psi} p_{\text{Beta}[\alpha_{j,a},\beta_{j,a}]}(U_{i,\ell})$, where $\alpha_{j,a} = (j-1)\psi + a$ and $\beta_{j,a} = \psi k + 1 - \alpha_{j,a}$. Using this setting will increase the sharpness of the splits, there by decreasing the regional overlap between the adjacent splits. As $\psi$ increases we combine more Beta distributions together, and this will lead to a more concentrated and refined edges of the distribution. In the limit as $\psi$ goes to infinity, we would recover the quantile splits—thus, this can be seen as relaxation of quantile-based splitting. For the second issue due to zero or near zero support, while our smoothed quantiles can alleviate this somewhat, the support may still be near-zero in certain regions. Thus there will be no samples in training corresponding to the held-out samples. We approach this by mixing our previously defined mixture of the Beta distributions with the uniform distribution as follows: $p(U_{i,\ell}|S_{i,\ell}) = (1-\epsilon)p_{\text{BetaMix}}(U_{i,\ell}|S_{i,\ell}) + \epsilon \cdot p_{\text{Unif}[0,1]}(U_{i,\ell})$ 

$$(1)$$

where $\epsilon \in [0, 1]$ is the mixing parameter. This addition will ensure a minimum representation of each region of support in our split, and if $\epsilon = 1$, we will get completely random splitting which is similar to standard CV splitting. We summarize our whole splitting procedure in Figure 2a, and illustrate the effects of different parameters on our splits in Figure 3. Additional figures for illustrating the Beta related splits are in Appendix C. We also prove Proposition 1, which states that this current choice of $p(U_{i,\ell}|S_{i,\ell})$ will yield biased splits, in Appendix F.

**Proposition 1.** *For any $\epsilon < 1$ and $\psi \in \{1, 2, \dots\}$ and assuming the splits are equal size in expectation, i.e., $p(S_{i,\ell}) = \frac{1}{k}$, if $p(U_{i,\ell}|S_{i,\ell}) = (1-\epsilon)p_{\text{BetaMix}}(U_{i,\ell}|S_{i,\ell}) + \epsilon p_{\text{Unif}[0,1]}(U_{i,\ell})$, where*

$$p_{\text{BetaMix}}(U_{i,\ell}|S_{i,\ell}=j) = \frac{1}{\psi} \sum_{a=1}^{\psi} p_{\text{Beta}[\alpha_{j,a},\beta_{j,a}]}(U_{i,\ell})$$

*and where $\alpha_{j,a} \triangleq (j - 1)\psi + a$ and $\beta_{j,a} \triangleq \psi k + 1 - \alpha_{j,a}$, then $p(U_{i,\ell}) = \text{Uniform}[0, 1]$ and the splits will be biased, i.e., $p(S_{i,\ell}|G_i) = p(S_{i,\ell}|U_{i,\ell}) \neq p(S_{i,\ell})$ or equivalently $I(S_{i,\ell}, G_i) > 0$.*

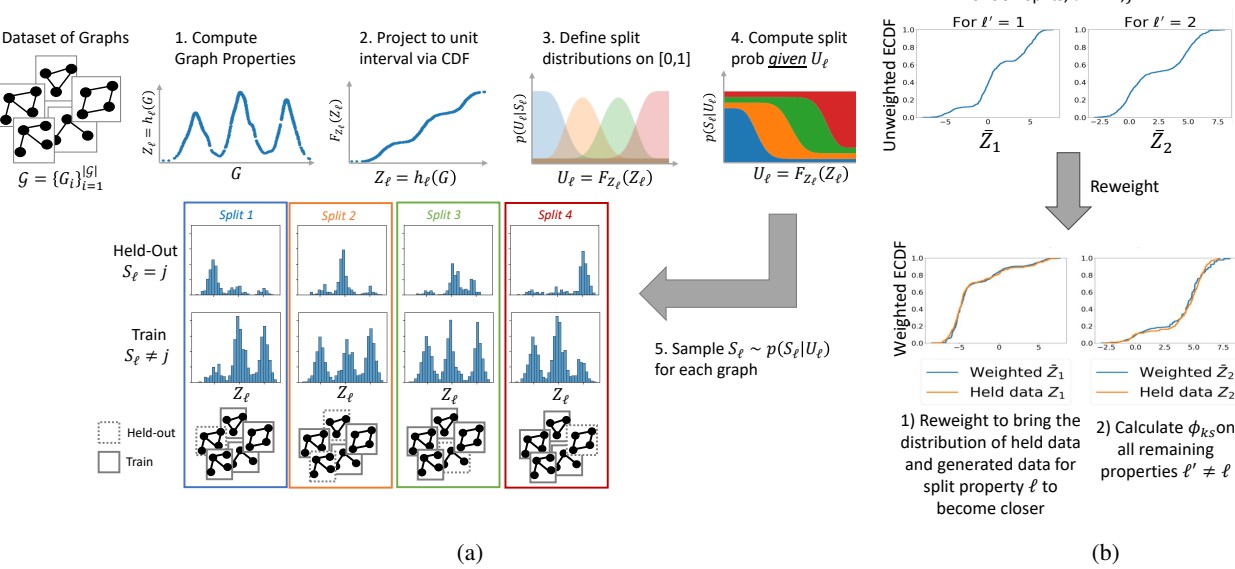

(a)

(b)

Figure 2: (a) The VV splitting process has 5 steps: 1) compute the relevant graph properties for each graph, 2) project samples via the CDF to a uniform distribution 3) Define the split distributions via a mixture of Beta distributions and a unifrom distribution, 4) Compute the split probability conditioned on $U_\ell$ using Bayes rule, and finally 5) sample the split variable based on these conditional probabilities. This will result in different splits. In the histograms above we plot the distribution of the split property $\ell$ in both the train and held parts for different splits. (b) An illustration of the reweighting process performed by VV for one of the splits (for $j = 1$ and $\ell = 1$ where the total number of properties is $m = 2$ ).

## 3.2 STEP 2: DEFINING A META-METRIC TO ADJUST FOR SHIFTED SPLITS

The second step in our approach is re-weighting the samples generated by our model to account for initially training the model with a biased dataset. The generative model ($\Omega$) was initially trained with $\mathcal{G}_{\text{train}}^{(\ell,j)}$ to produce samples $\mathcal{G}_{\text{gen}}^{(\ell,j)}$ (whose true distribution is $q(\bar{G}^{(\ell,j)}|\theta = \theta^*_{\Omega(\mathcal{G}_{\text{train}}^{(\ell,j)})})$). Those generated samples -or more precisely the properties of such generated samples which we can denote by $\bar{Z}$- will follow a similar distribution to that of the training dataset (assuming that the model doesn't underfit), but will be different from the distribution of the held-out samples $\mathcal{G}_{\text{held}}^{(\ell,j)}$. We will refer to the true distributions of a property $\ell$ in $\mathcal{G}_{\text{train}}^{(\ell,j)}$ and $\mathcal{G}_{\text{held}}^{(\ell,j)}$ as $p(Z_{i,\ell}|S_{i,\ell} \neq j)$ and $p(Z_{i,\ell}|S_{i,\ell} = j)$ respectively, and to that of the generated samples as $q(\bar{Z}_{i,\ell}|\theta = \theta^*_{\Omega(\mathcal{G}_{\text{train}}^{(\ell,j)})})$.

To account for such a shift in distribution, we need to un-bias the generated samples, and one way of doing so is by re-weighting them, thus we define the importance weight of each sample $\bar{G}_i$ in $\mathcal{G}_{\text{gen}}^{(\ell,j)}$ with property $\bar{Z}_{i,\ell}$ as follows: $W^{(\ell,j)}(\bar{G}_i) :=$

$$W^{(\ell,j)}(\bar{Z}_i = h(\bar{G}_i)) := \frac{p(\bar{Z}_{i,\ell}|S_{i,\ell} = j)}{q(\bar{Z}_{i,\ell}|\theta = \theta^*_{\Omega(\mathcal{G}_{\text{train}}^{(\ell,j)})})} \quad (2)$$

Using the definition above, we can extend our graph multi sets to a weighted version where $\mathcal{G}_W = \{\!\{(G_i, W(G_i)) : G_i \in \mathcal{G}\}\!\}$, so for $\mathcal{G}_{\text{gen}}$ we will have a corresponding $\mathcal{G}_{\text{gen},W}$. Furthermore, let $\phi$ be a metric that can compare two distributions and handle weighted samples, we can define our meta-metric as:

$$\phi_{\text{VV}}(\mathcal{G}_{\text{held},\mathbf{1}}^{(\ell,j)}, \mathcal{G}_{\text{gen},W}^{(\ell,j)}; \phi) = \phi(\mathcal{G}_{\text{held},\mathbf{1}}^{(\ell,j)}, \mathcal{G}_{\text{gen},W}^{(\ell,j)}) \quad (3)$$

For two-sample tests, we will use an estimate of the number of effective samples based on [Monahan, 2011, Sec. 12.4] defined as:

$$N_{\text{eff}}(\mathcal{G}_{\text{gen},W}^{(\ell,j)}) = \frac{(\sum_{(\bar{G}_i, W(\bar{G}_i)) \in \mathcal{G}_{\text{gen},W}^{(\ell,j)}} W^{(\ell,j)}(\bar{G}_i))^2}{\sum_{(\bar{G}_i, W(\bar{G}_i)) \in \mathcal{G}_{\text{gen},W}^{(\ell,j)}} W^{(\ell,j)}(\bar{G}_i)^2}. \quad (4)$$

### 3.2.1 Concrete Instantiation of the Metric via KS Statistics

As one instantiation for our generic framework, we can choose $\phi$ to be a weighted version of a two-sample KS statistic. One of the reasons for choosing the KS statistic is that its values are always between 0 and 1, and its weighted version can handle weighted samples from two distributions. Given any two weighted graph datasets $\mathcal{G}_{1,W_1}$ and $\mathcal{G}_{2,W_2}$. The empirical weighted KS statistic of a specific graph property function $h(\cdot)$ with $Z_i = h(G_i)$ can be defined as follows:

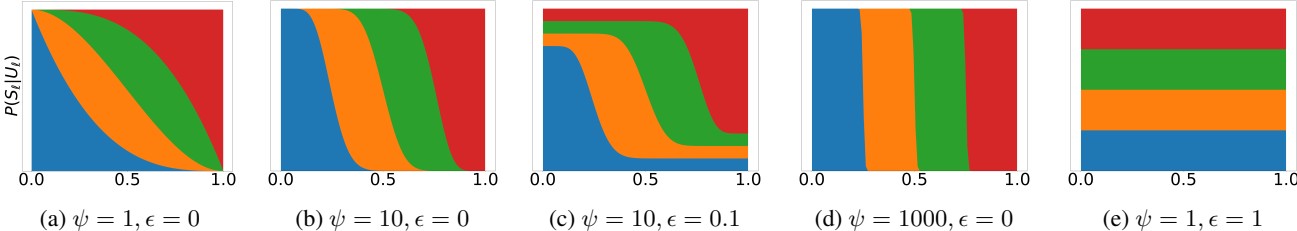

| (a) $\psi = 1, \epsilon = 0$ | (b) $\psi = 10, \epsilon = 0$ | (c) $\psi = 10, \epsilon = 0.1$ | (d) $\psi = 1000, \epsilon = 0$ | (e) $\psi = 1, \epsilon = 1$ |

Figure 3: The Figure showcases stacked conditional split probabilities $p(S_\ell | U_\ell)$ obtained using VV with different sharpness $\psi$ and uniform mixing parameter $\epsilon$. In (a), the split distributions may overlap too much when $\psi = 1$. In (b), the splits are sharper but still smooth. In (c), the uniform mixing parameter allows all splits to have some support. In (d) and (e), we demonstrate the extremes of our approach that yield either quantile splits as $\psi \to \infty$ or uniform splits as in standard CV if $\epsilon = 1$.

$$\phi_{\mathrm{KS}}(\mathcal{G}_{1,W_1}, \mathcal{G}_{2,W_2}; h) = \sup_{G_i} |\hat{F}_{W1}(h(G_i)) - \hat{F}_{W2}(h(G_i))|$$

$$= \sup_{Z_i} |\hat{F}_{W1}(Z_i) - \hat{F}_{W2}(Z_i)|, \qquad (5)$$

where $\hat{F}_{W1}(z) = \frac{1}{\sum_i W_i} \sum_i W_i \mathbf{1}_{(Z_i \le z)}$ is the weighted empirical CDF associated with $\mathcal{G}_{1,W_1}$ and similarly for $\hat{F}_{W2}(z)$ associated with $\mathcal{G}_{2,W_2}$. For our specific case, we want to obtain that metric between our unweighted $\mathcal{G}_{\mathrm{held},1}^{(\ell,j)}$ dataset (we can assume it's re-weighted by ones) and the re-weighted generated samples $\mathcal{G}_{\mathrm{gen},W}^{(\ell,j)}$ with the chosen graph property function $h_{\ell'}$ we would like to evaluate on, therefore we'll have:

$$\phi_{\mathrm{VV}}(\mathcal{G}_{\mathrm{held}}^{(\ell,j)}, \mathcal{G}_{\mathrm{gen}}^{(\ell,j)}; \phi_{\mathrm{KS}}, h_{\ell'}) = \phi_{\mathrm{KS}}(\mathcal{G}_{\mathrm{held},1}^{(\ell,j)}, \mathcal{G}_{\mathrm{gen},W}^{(\ell,j)}; h_{\ell'}), \qquad (6)$$

where $\ell' \in \{1, .., m\}$ and where $\ell' \ne \ell$. For an illustration of our unbiasing and re weighting procedure refer to Figure 2b (A full version of the figure is presented in Appendix B). We also prove that this chosen metric will converge to zero if the model generalizes well with the rate of convergence depending on the number of samples in the dataset. Below we state the theorem, while the full proof is presented in Appendix F.

**Theorem 1** ($\phi_{\mathrm{KS}}(\mathcal{G}_{\mathrm{held},1}^{(\ell,j)}, \mathcal{G}_{\mathrm{gen},W}^{(\ell,j)}; h_{\ell'})$ consistent). *Using VV for generating data splits and corresponding datasets $\mathcal{G}_{\mathrm{train}}^{(\ell,j)}$, $\mathcal{G}_{\mathrm{held}}^{(\ell,j)}$ and using an implicit generator $\Omega$ trained on $\mathcal{G}_{\mathrm{train}}^{(\ell,j)}$ to generate data $\mathcal{G}_{\mathrm{gen}}^{(\ell,j)}$.*

*Then, if $\mathcal{G}_{\mathrm{gen}}^{(\ell,j)}$ is generated with the same distribution as $\mathcal{G}_{\mathrm{held}}^{(\ell,j)}$, for any $\epsilon \in [0, 1]$,*

$$P(\phi_{KS}(\mathcal{G}_{\mathrm{held},1}^{(\ell,j)}, \mathcal{G}_{\mathrm{gen},W}^{(\ell,j)}; h_{\ell'}) > \epsilon) \le$$

$$4 \exp\left(-2 \min(|\mathcal{G}_{\mathrm{gen}}^{(\ell,j)}|, |\mathcal{G}_{\mathrm{held}}^{(\ell,j)}|)\left(\frac{\epsilon}{2}\right)^2\right), \qquad (7)$$

**Implementation of Vertical Validation** The comprehensive implementation details are presented in Appendix E, but we summarize a few key points here. First, we use a

smoothed version of the empirical CDF instead of the true CDF of graph properties. Second, we Kernel Mean Matching (KMM) [Huang et al., 2006] to estimate the importance weights. Lastly, to sample from the generative model, we iteratively generate samples until the count of effective samples reaches a predetermined fixed number.

## 4 EXPERIMENTS

In our experiments, we first demonstrate the effectiveness of our method in measuring the generalization of graph generative models. Once this foundation is established, we proceed to compare different graph generative models using our approach.

### 4.1 VALIDATING THE EFFECTIVENESS OF VV

For the experiments in this section, we focus on one split property $\ell$ and one split index $j$ in VV, similar to concentrating on a single fold in the traditional CV scheme, which we will denote as HV for Horizontal Validation. We also opt to use data with a known distribution, providing access to ground truth for generating additional samples when necessary. We generate 500 Erdős-Rényi random graphs [Erdős and Rényi, 1960] with $n_{\mathrm{nodes}} = 20$ and $p = 0.5$. Additionally, we generate 500 samples of the community dataset (which we refer to as Comm20), consisting of two equally sized communities with $n_{\mathrm{nodes}}$ ranging between 6 and 10 per community (12-20 in total). The probability of an edge within each community is $p = 0.7$, while across communities, it's a function of the number of nodes within a community, $p_{int} = \frac{0.1}{n_{\mathrm{nodes}/2}}$.

In our experiments, we explore five possible model types: 1) **E.Memo**, Exact Memorization, represents a model that memorizes the original training data, then sample "new" graphs only by bootstrapping from it. 2) **A.Memo**, Approximate Memorization, represents a model that memorizes the training data but introduces subtle variations during sampling by adding or removing an edge from graphs in

the memorized data. 3) **Oracle**, represents a model that is capable of generating data directly from the ground truth distribution (For Erdős-Rényi graphs the parameters of such model is $n_{\text{node}} = 20$ and $p = 0.5$, for Comm20, the parameters of such model is $n_{\text{node}} = 12 - 20$ and $p = 0.7$), which serves as the benchmark for the ideal generative model; 4) **Close**, representing a model that posses the ability to generate data from a distribution that is somewhat "close" to the ground truth distribution (generating new graphs for Erdős-Rényi with $n_{\text{node}} = 20$ and $p = 0.45$, and for Comm20 with $n_{\text{node}} = 12 - 20$ and $p = 0.65$). 5) **Far**, representing a model that simulates under fitting as it generates data from a distribution that is not close enough to the ground truth (generating new graphs for Erdős-Rényi with $n_{\text{node}} = 20$ and $p = 0.4$, and for Comm20 with $n_{\text{node}} = 12 - 20$ and $p = 0.6$).

To illustrate the usefulness of our VV approach in model selection, we use VV with $\epsilon = 0.01$, $\ell = 2$ (corresponding to the split property: number of triads), $\psi = 10$ and $k = 5$ to create a train-test split. We choose to hold out the last split corresponding to $j = 5$ and we will refer to that part as v-test and to the parts corresponding to $j = 1, 2, 3, 4$ as train. Next we use VV again to further split the train part. In this case we use $\epsilon = 0.01$, $\ell = 2$, $\psi = 10$ and $k = 4$ to create a train-val split. We again choose to hold out the last split corresponding to $j = 4$ and will refer to it as v-val, while the splits $j = 1, 2, 3$ are referred to as v-train. Then in an alternate setting, we use HV (which we can achieve by adjusting our model parameters to $\epsilon = 1$ and $\psi = 1$) also on the train part to create train-val splits or folds, we assign the label of h-val to one of those folds (since they are all similar) and the rest we label h-train. The general idea is that v-test is an area we are interested in but don't have access to, traditionally we would use HV and splits like h-train/h-val to choose the best model. However we argue that this approach isn't ideal if the goal is having a model capable of generalizing to areas of thin support, and that using a split like v-train/v-val can help us choose the best model for that task. To illustrate this, we train the five models that we previously introduced on Erdos-Renyi and Comm20 datasets, each of the models is trained and tested on train/v-test, v-train/v-val and h-train/h-val (those splits are illustrated in Figure 12 in Appendix D), we then report the averaged KS for testing on the average degree ($\phi_{ks}^D$) and the average clustering coefficient ($\phi_{ks}^C$) properties for all models and split types in Figure 4, where we see some trends. First, in the results on h-val, we see that E.MEMO and A.MEMO seem to have lower KS values than oracle, this can be explained by the fact that both these models care only about in-distribution performance, and as such can achieve low results compared to the oracle that should have been the best model if we hadn't initially held out v-test ie. the oracle produced data covering all of the support, but our h-val only covers part of that support and hence HV viewed the performance as inferior. Second, we notice

that the oracle for both of our cases (v-val and v-test) is indeed regarded as the superior model since our goal is to give higher rankings to models that generalizes better. Third, as a side effect of our setting we were also able to detect that E.MEMO and A.MEMO are models that are inferior to oracle since they won't be able to generalize. To summarize, if a user was interested in a model capable of generalizing to regions of thin support, then the conventional HV setting is misleading in model selection, while a VV setting is favourable.

## 4.2 COMPARING MODELS WITH VV

### 4.2.1 Using VV in a Train-Val-Test context

Building on the empirical validation of our metric presented in the previous section, we now proceed to compare real graph generative models on realistic datasets using our VV method. We have implemented our approach in a manner consistent with the validation experiments described in section 4.1. However, in this section, the different models correspond to various representative graph generative models.

**Models and Datasets**   We select two representative graph generative models for comparison: DiGress [Vignac et al., 2022], a discrete diffusion-based model that introduces noise to graphs and then trains a graph transformer to revert the process, and GDSS [Jo et al., 2022], a score-based diffusion model employing stochastic differential equations (SDEs) to generate node, edge attributes, and adjacency matrices jointly. For this experiment, we choose Qm9, a commonly used molecular dataset of 130,831 small molecules. We selected five properitie (i.e., $m = 5$) for Qm9: average degree, molecular weight (Mlwt), Topological Polar Surface Area (TPSA) [Prasanna and Doerksen, 2009], ring counts, and the logarithm of the partition coefficient (logp). We preprocessed the dataset by removing hydrogen atoms and filtering out molecules where any of the five properties could not be calculated using the rdkit package. Additional experimental details are outlined in Appendix G.

**Model Comparison Experiment**   We aim at judging the performance of different generative models by their ability to generalize. To accomplish this, we use TPSA as the split property (which corresponds to index $\ell = 3$) with parameters $\psi = 10$, $\epsilon = 0.01$, $k = 5$ and hold out the split corresponding to $j = 5$ as the v-test portion. Then, we further split the data with $k = 4$ and hold out the split corresponding to $j = 4$ as the v-val portion. The remaining splits $j = 1, 2, 3$ correspond to v-train and are used for training the model. We then evaluate the performance of these models with respect to v-val and v-test for each model type by generating samples from the trained models such that the effective number of samples is $n_{\text{eff}} = \min(1000, |\mathcal{G}_{\text{held}}^{(\ell, j)}|)$. Finally, we calculate the $\phi_{KS}$ scores on the five properties

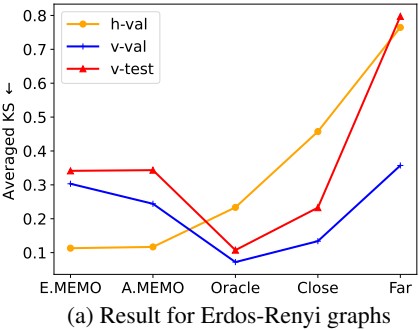

(a) Result for Erdos-Renyi graphs

(b) Results for Comm20 graphs

Figure 4: On both Erdos-Renyi and Comm20 datasets, our proposed vertical validation approach (VV) can select the best model for generating in thin support as shown by the test line, whereas the standard train-test splitting (CV) tends to favor memorization despite poor generalization to the thin support regions. Note that for CV, the oracle distribution seems worse than memorization because oracle is generating from the true distribution rather than the shifted training distribution—thus it appears to CV that memorizing is actually a better option. This phenomena does not happen in our validation approach because we aim to find a model that generalizes to the thin support well. This also showcases that our approach is better able to detect memorization than the standard train-test split validation.

Table 1: $\phi_{KS}$ values for different test properties for Qm9 when compared against v-val and v-test where $\phi_{ks}^{D}$, $\phi_{ks}^{Mlwt}$, $\phi_{ks}^{RC}$, $\phi_{ks}^{LogP}$, $\phi_{ks}^{Avg}$ are average degree, molecular weight, average ring counts, average logP, and the total average over all these properties respectively

| | Model | $\phi_{ks}^{D}$ | $\phi_{ks}^{Mlwt}$ | $\phi_{ks}^{RC}$ | $\phi_{ks}^{LogP}$ | $\phi_{ks}^{Avg}$ |
|---|---|---|---|---|---|---|
| vval | DiGress | 0.206 | 0.531 | 0.206 | **0.048** | 0.248 |
| | GDSS | **0.053** | **0.343** | **0.053** | 0.083 | **0.133** |
| vtest | DiGress | 0.216 | 0.568 | 0.204 | **0.109** | 0.274 |
| | GDSS | **0.058** | **0.444** | **0.058** | 0.123 | **0.171** |

of interest excluding the cases when the test property is the same as the split property ($\ell' \neq \ell$).

We see in Table 1 that our VV method using v-val is able to correctly select the model which will perform best on thin support, i.e., perform the best on the held-out v-test. In most cases, GDSS is better on both v-val and v-test, but in the case of LogP, DiGress is better on both v-val and v-test.

Both models seem to struggle when it comes to molecular weight suggesting the inherent difficulty of generalizing over that property. This result showcases that using VV can properly select between model classes when generalization on thin support is desired—and this selection may depend on the test property.

**Exploratory Visualizations:**   To explore the properties of our metric better, we used the samples generated from the experiment above, sorted them in descending order according to the weights assigned by our approach, then filtered for validity and novelty to get the top weighted 100 molecules. We then visualized the top four of these molecules generated by DiGress when testing against v-test in Figure 5 and visualized the rest of the top generated molecules in Ap-

pendix G. Furthermore, in Figure 6, we indicate the value of the split property (TPSA) of those 4 molecules and their location with respect to the entire distribution of the TPSA property. As expected, the samples with the higher weights tend to be from the region that the data was held from.

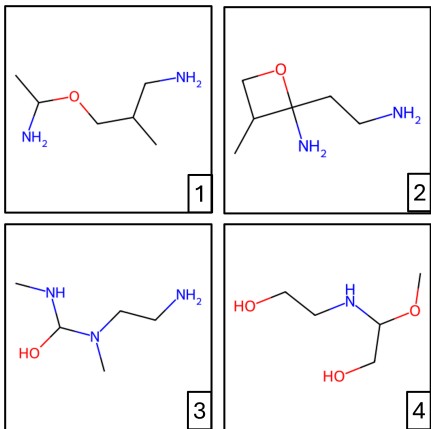

Figure 5: Example of the generated molecules from DiGress. These are the top 4 -after filtering for validity and novelty-according to the weights assigned by our method when using v-test as the held out portion.

### 4.2.2   Using VV in a Cross-Validation-Like Context

In this section, we choose to train exhaustively on all the splits from the corresponding split properties and split indices. This approach is similar in spirit to cross-validation, but rather than having $k$ folds only, we will have $k \times m$ folds, corresponding to each split index $j \in 1, .., k$ and split feature $\ell \in 1, ..., m$. For our experiments, we chose $k = 4$ and $m = 5$ resulting in a total of 20 different data splits. We

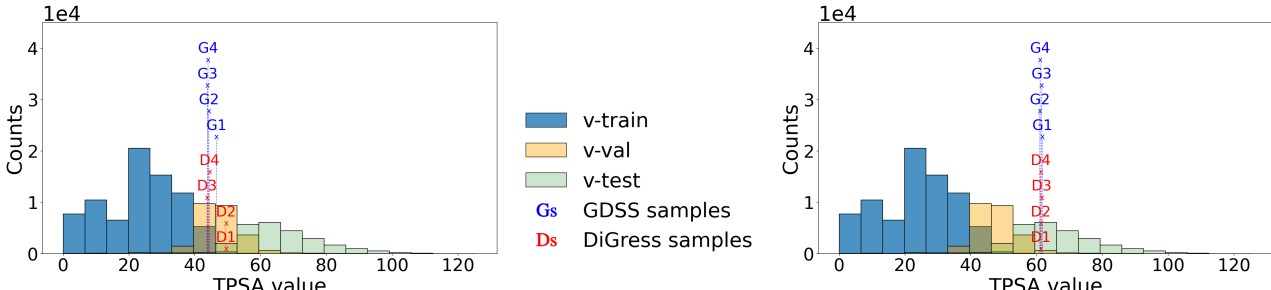

Figure 6: Both figures show the distribution of the TPSA property for v-train/v-val/v-test. As expected the top four highest weighted molecules after filtering for validity and novelty are in high density regions of v-val (left) and v-test (right). The distribution of v-train is in blue and is overlayed with the distribution of the v-val portion in yellow and v-test in green. We plotted where would our top molecules lie with respect to their calculated TPSA value (the value of the y-axis is not meaningful for the molecules, we varied it across the molecule for ease of visualization). We use the first letter to signify which model generated the sample D for DiGress and G for GDSS.

trained each model on our splits separately and generated samples from these trained models such that the effective number of samples is $n_{\text{eff}} = min(1000, |\mathcal{G}_{\text{held}}^{(\ell,j)}|)$. Finally, we evaluated the performance of these model on the 5 properties of interest excluding the cases when the test property is the same as the split property ($\ell' \neq \ell$).

In addition to the representative models mentioned in section 4.2.1, we also include an additional model GGAN [Krawczuk et al., 2021], which is a GAN-based model that employs adversarial training to generate graphs. This model is suitable only for non molecular datasets as node properties are not easily incorporated. Thus we choose to evaluate all 3 models (DiGress, GDSS and GGAN) on a variation of the previously used community dataset which we descibe in more details in Appendix G and refer to as Comm dataset, and we choose to evaluate DiGress and GDSS on Qm9 again in this current context. We elaborate more on the results of Comm dataset below, and also present the results on Qm9 in Table 3 for completeness. For the results on the Comm dataset, We aggregate according to different use cases depending on the user's interest.

*Use Case 1*: The user seeks the best model for overall performance across all predefined $m$ properties: we calculate the average $\phi_{KS}$ across all combinations of $\ell$, $j$, and $\ell'$, with $\ell \neq \ell'$, and we report a single averaged value per model. Lower values indicating better generalization capabilities. To demonstrate this case, we report the overall average performance for: DiGress: 0.567 GDSS: 0.51 and GGAN: 0.17. From these numbers, it seems that GGAN performed the best, followed by GDSS, with DiGress falling slightly behind. It can also be useful to consider more specific use cases rather than this broad one.

*Use Case 2*: The user aims to identify a model that excels in generalizing over a specific property of interest: we can compute separate average $\phi_{KS}$ values for each test property $\ell'$ by averaging across all ($\ell,j$) splits s.t $\ell' \neq \ell$. This will enable the user to make decisions based on the performance

Table 2: Average $\phi_{KS}$ values for different test properties in Use Cases 2 and 3: $\phi_{ks}^{D}, \phi_{ks}^{T}, \phi_{ks}^{S}, \phi_{ks}^{C}, \phi_{ks}^{M}$ are average degree, average number of triads, average shortest path length, average clustering coefficient, average maximal cliques, respectively

|  | Model | $\phi_{ks}^{D}$ | $\phi_{ks}^{T}$ | $\phi_{ks}^{S}$ | $\phi_{ks}^{C}$ | $\phi_{ks}^{M}$ |
|---|---|---|---|---|---|---|
| Use Case 2 | DiGress | 0.678 | 0.59 | 0.779 | 0.493 | 0.333 |
| | GDSS | 0.546 | 0.511 | 0.727 | 0.484 | 0.285 |
| | GGAN | **0.334** | **0.108** | **0.188** | **0.165** | **0.057** |
| Use Case 3 | DiGress | 0.688 | 0.601 | 0.783 | 0.437 | 0.294 |
| | GDSS | 0.516 | 0.47 | 0.709 | 0.492 | 0.293 |
| | GGAN | **0.376** | **0.06** | **0.155** | **0.167** | **0.044** |

of their specific property of interest. We report the results of this use case in Table 2. Based on the results, GGAN generally outperforms the other models. GDSS occupies middle ground, DiGress consistently falls behind. To further understand this behaviour, We examined the ECDFs of the generated properties for the models, and compared them to the ECDFs of the held-out data (see Appendix G for a list of Figures), and noticed an overall trend where the original data (held-out) tend to have pronounced discontinuities, a characteristic which GGAN tend to replicate with fewer modes. In contrast, models such as DiGress and GDSS demonstrate smoother distributions. Additionally, it appears that average shortest path length is the most challenging property for GDSS and DiGress models to capture and generalize correctly. Through examining the ECDFs we conjecture that GDSS and DiGress are not able to concentrate the generations near the middle of the distribution. Given their reliance on diffusion-based mechanisms, this observation could imply that while these models may perform adequately at higher noise levels, they exhibit diminished precision at lower noise levels.

*Use Case 3*: The user seeks a model that generalizes well on the edges of the distribution for a particular property: we can compute the average $\phi_{KS}$ of test properties $\ell'$ across all combinations of $\ell$ values (with $\ell' \neq \ell$) and for only $j = 1$

and $j = 4$ (since these particular splits are focused on the edges of the distribution, as illustrated in Figure 2a). The results are presented in Table 2 and are also consistent with Use Case 2, and they reveal distinct strengths and weaknesses in capturing various properties. GGAN consistently exhibits strong performance across most of the properties, while GDSS remains at second place. DiGress, although performing slightly better than GDSS in modeling average clustering coefficients and having a similar performance to GDSS in modeling maximal cliques, tends to rank lower overall.

For Qm9 dataset, the scores for use case 1 for DiGress were: 0.174, and for GDSS were: 0.096. The results of use cases 2 and 3 are presented in Table 3. Overall the results follow a similar trajectory to these of the Comm dataset in this setting, that is the performance of GDSS and DiGress were close, but GDSS overall achieves a better score. It is also worth comparing the results previously introduced in section 4.2.1 in Table 1 (and in particular those under v-val) to the current results of use case 2. We see that overall the trend didn't change, however the scores on the Mlwt property got better in the later suggesting that the particular split (i.e. the combination of the choice of split property $\ell$ and split index $j$ ) chosen in section 4.2.1 was a particularly hard one, as averaging over multiple splits made the scores better. This again emphasize the role of choosing the split features, and we discuss this point in more details in Appendix G and in section 5

Table 3: Average $\phi_{KS}$ values for different test properties for Use Cases 2 and 3: $\phi_{ks}^{\text{D}}, \phi_{ks}^{\text{Mlwt}}, \phi_{ks}^{\text{TPSA}}, \phi_{ks}^{\text{RC}}, \phi_{ks}^{\text{LogP}}$ are average degree, molecular weight, TPSA, average Ring Counts, and logp

| | Model | $\phi_{ks}^{\text{D}}$ | $\phi_{ks}^{\text{Mlwt}}$ | $\phi_{ks}^{\text{TPSA}}$ | $\phi_{ks}^{\text{RC}}$ | $\phi_{ks}^{\text{LogP}}$ |
|---|---|---|---|---|---|---|
| Use Case 2 | DiGress | 0.136 | 0.368 | 0.082 | 0.128 | 0.156 |
| | GDSS | **0.088** | **0.171** | **0.080** | **0.080** | **0.059** |
| Use Case 3 | DiGress | 0.113 | 0.347 | 0.067 | 0.102 | 0.154 |
| | GDSS | **0.100** | **0.173** | **0.067** | **0.086** | **0.070** |

## 5 DISCUSSION AND CONCLUSION

**Generating Molecules on Thin Support Regions** In practice, novel molecule generation may focus on generating molecules within the thick support (rather than thin support) of the *marginal* property distributions because those molecules would be most similar to known molecules. However, we argue that evaluating generation on thin support is still important because thin support regions in the *joint* distribution could be hidden in the thick support of *marginal* distributions, especially when considering a high-dimensional distributions. For example, consider samples on a 3D sphere. When projected onto any of the three dimensions, it will look like the support is dense near zero. However, the distribution has no support at or near the all zero vector. Thus,

we hypothesize that in high dimensional spaces, there are many thin support regions that are hidden. When we systematically create thin support regions using our approach, the goal is to measure the model's ability to generalize to thin support in general (including thin support of the *joint* distribution). Thus, while in practice novel molecule generation may focus on generating molecules with the thick support of the marginal property distributions, we test the ability of the model to generate in those regions as this will reflect its ability to generate in thin support of the *joint* distribution.

**Limitations** Our approach when used exhaustively as presented in the experiments of section 4.2.2 can be computationally burdensome, however practically we would choose only a single split feature and a single split index as we did in section 4.2.1 and this would avoid the added computational cost. While we recommend choosing a split feature that is maximally dependent on other features as we discuss in Appendix G, choosing the split property is still an area of potential optimization. Additionally, because our method depends heavily on the joint distribution of the chosen properties, we recommend that the user pre-examine the property distributions and carefully select relevant properties, where properties with smooth distributions will likely be better for evaluation. Also, our method is limited to 1-dimensional properties. Generalizing our method to multivariate splits is an area for future work. Finally, we note that estimating sample weights is complex and while our kernel mean matching (KMM) approach worked reasonably well in our case, choosing the kernel parameters or using more advanced weight estimation approaches is an open area of exploration (more discussion in Appendix E). Therefore, we hope our work opens up new avenues of research

**Conclusion** In summary, we introduced Vertical Validation, a new framework for biased splitting and reweighting, to evaluate the generalizability of implicit graph generative models on thin support regions. We developed a practical algorithm to perform this given a set of graph properties. We demonstrated that this validation approach can be used to select models which will generalize better to thin support regions. Ultimately, we hope that our approach is a step in establishing more concrete and robust evaluation methodologies for graph generative models.

## ACKNOWLEDGEMENTS

M.E. and D.I. acknowledge support from ARL (W911NF-2020-221). This work was funded in part by the National Science Foundation (NSF) awards, CCF-1918483, CAREER IIS-1943364 and CNS-2212160, Amazon Research Award, AnalytiXIN, and the Wabash Heartland Innovation Network (WHIN), Ford, NVidia, CISCO, and Amazon. Any opinions, findings and conclusions or recommendations expressed in this material are those of the authors and do not necessarily reflect the views of the sponsors.

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

# Appendix

## Table of Contents

## A   MORE DETAILS ON THE PROBLEMS OF MEAN BASED METHODS

We illustrate the problem of man based methods in Figure 7 where we show that using cross-validation with the difference in means or the more complex Wasserstein-1 distance[2] does not provide useful signal for selecting a model whereas negative log-likelihood provides a strong signal. In summary, prior evaluation approaches for implicit generative models are limited in their ability to measure performance on thin support regions. In Figure 7, we also show that our VV approach provides reliable signal for selecting the correct model in this toy example.

## B   DETAILED ILLUSTRATION OF VV'S REWEIGHING PROCESS

In the main paper in Figure 2a, we presented a figure that illustrates the splitting steps of our approach, and here in Figure 8 we illustrate the complete reweighting process of our VV approach. In this illustration, we focus on a particular split and split feature (for $j = 1$ and $\ell = 2$) and where the total number of properties is $m = 2$.

## C   ILLUSTRATION OF BETA SPLITS

In section 3.1 we presented the Beta Splits, and here we add more figures that illustrate how Beta-based splitting works with different parameters. Figure 9 shows an example for $k = 4$ beta distributions (and $\psi = 1$) and Figure 10 shows the same example but with sharpness of 10, i.e., $\psi = 10$. In Figure 11, we show an illustration of using the uniform mixing parameter $\epsilon$ to ensure some support on the whole range of the uniform.

---

[2]Wasserstein-1 distance is also an integral probability metric like MMD but uses a different class of functions in the optimization problem. Wasserstein-1 was chosen for this illustration because it can be computed efficiently and has no hyperparameters.

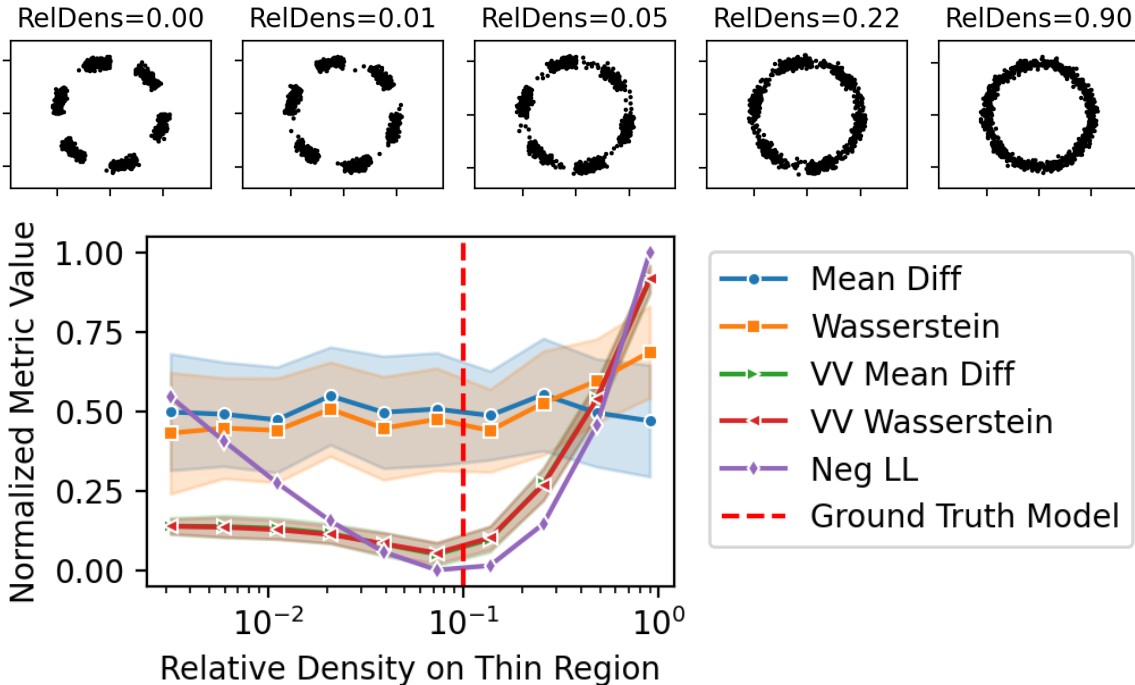

Figure 7: While using standard cross-validation with mean difference or Wasserstein-1 metrics does not provide reliable signal to select the right model, our vertical validation (VV) method with mean difference or Wasserstein-1 provides reliable information on the best model and matches the ranking of the negative log-likelihood. This illustration uses the 2D dataset from Figure 1 as ground truth, and the relative density of the "thin region" is varied to represent different estimated models (top). For both standard validation and vertical validation, we use 10 folds and 30 repetitions and show the standard deviation for each method.

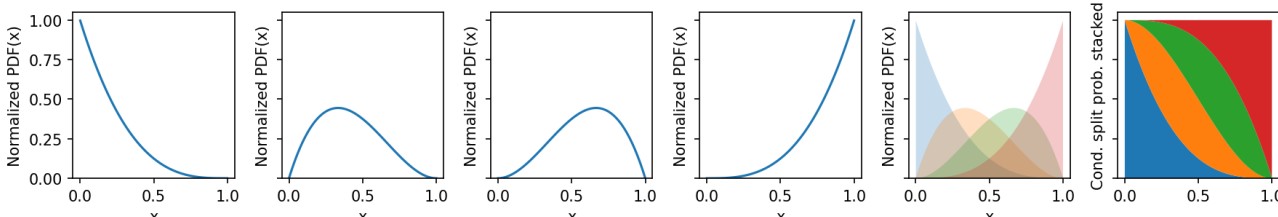

Figure 9: First 4 figures are the normalized PDF for 4 Beta distributions parameterized by a set of shifting parameters. The fifth figure is the aggregation of the 4 figures to the left, and the right most figure is the conditional probabilities of a projected point falling in a split based on the previously described Beta distributions where each split have a different color.

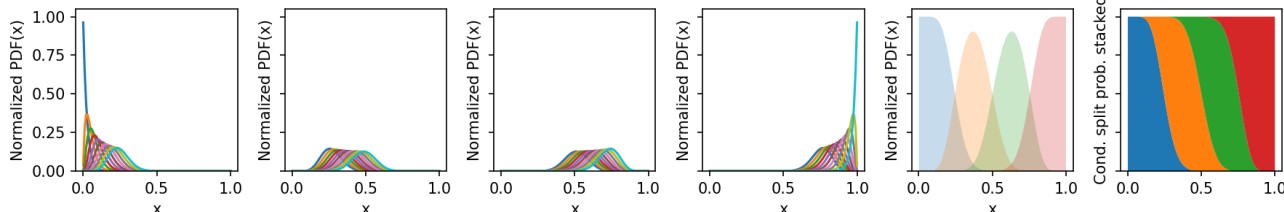

Figure 10: The same plot as Figure 9 except where the sharpness is 10, i.e. $\psi = 10$. We notice now that the conditional probabilities computed have sharper edges as seen in the rightmost figure.

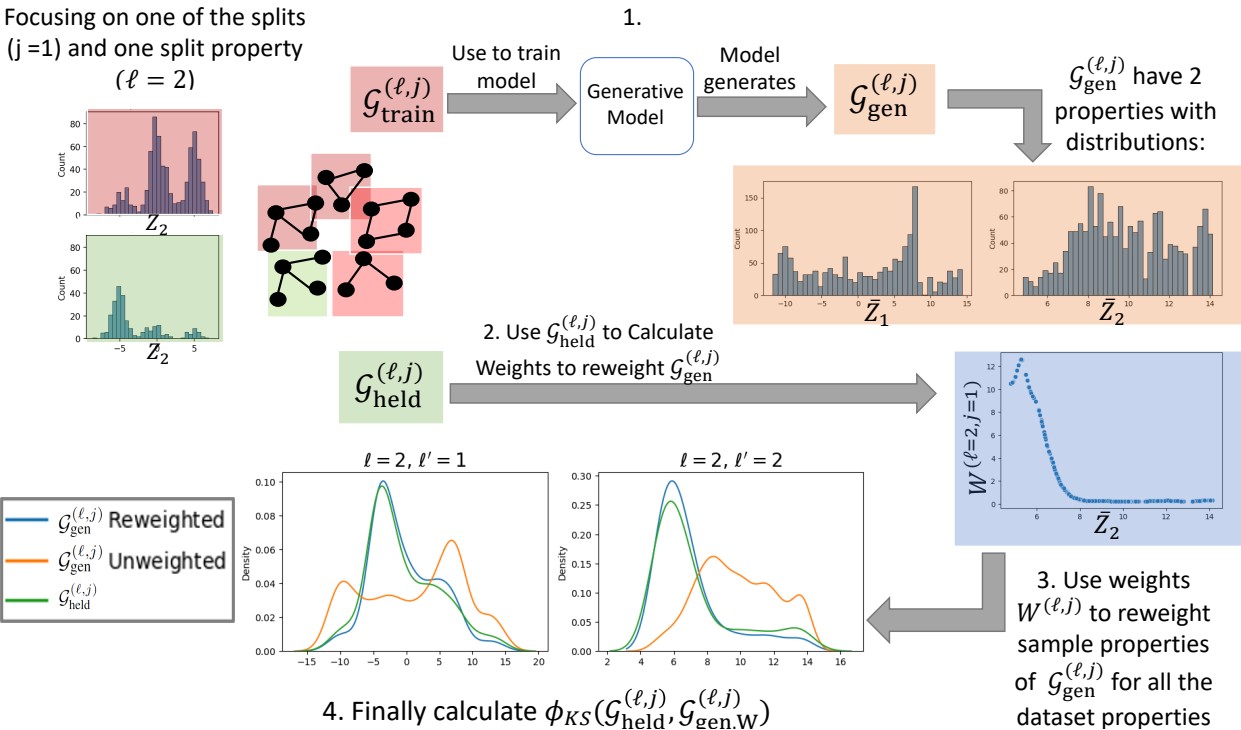

Figure 8: (1) using $\mathcal{G}_{\text{train}}^{(\ell,j)}$ we train a generative model which then generates $\mathcal{G}_{\text{gen}}^{(\ell,j)}$. For the generated dataset we can compute the properties of interest ($\bar{Z}_1$ and $\bar{Z}_2$). (2) Use $\mathcal{G}_{\text{held}}^{(\ell,j)}$ to calculate the weights $W^{(\ell,j)}$ which will then be used in (3) to re-weight the samples in $\mathcal{G}_{\text{gen}}^{(\ell,j)}$ by applying those weights to the calculated properties. The Density figures above show the difference in distribution between the re-weighted and un-weighted versions of the generated data as well as how they compare to the held-out part of the data. (4) Finally we can compute the KS statistic between the weighted $\mathcal{G}_{\text{gen}}^{(\ell,j)}$ and $\mathcal{G}_{\text{held}}^{(\ell,j)}$ as detailed in Equation 6.

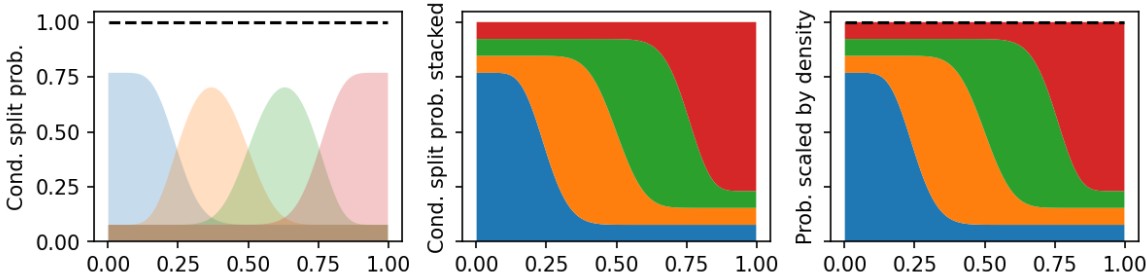

Figure 11: Leftmost is the mixture of betas for the different splits along with the uniform $\epsilon = 0.1$ (and using $\psi = 10$). Middle is the stacked probabilities. Rightmost are the probabilities normalized by the density.

# D ILLUSTRATION OF THE SPLITTING APPROACH DESCRIBED IN SECTION 4.1

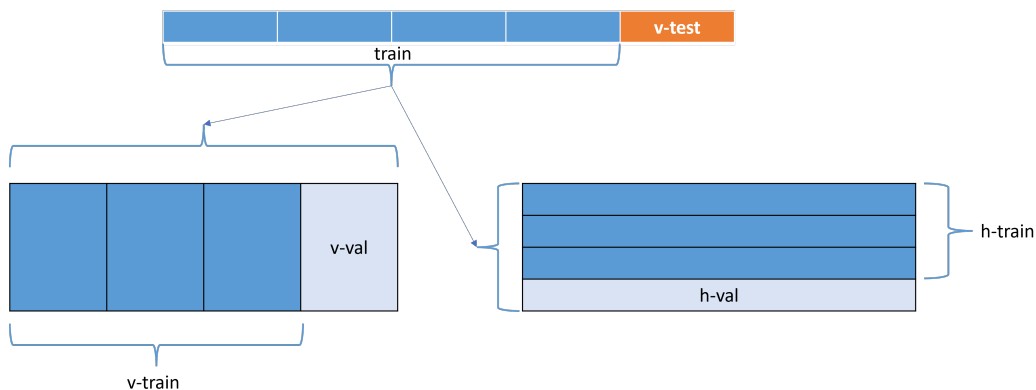

Figure 12: An illustration demonstrating the created splits described in the main section

# E IMPLEMENTATION DETAILS OF VV

We discuss a few implementation details here including how to project samples to the unit interval via the empirical CDF, how to estimate importance weights via kernel mean matching, and how to generate enough samples from the model (as they depend on the importance weights).

## E.1 A GENERALIZATION OF THE EMPIRICAL CDF

To project samples onto the unit interval, we use a generalization of the empirical CDF (ECDF) where the test points may be different from the training points and where ties (e.g., in discrete spaces) can be handled appropriately. The core idea is that for test points we want to project, denoted by $\mathcal{A}$, we find the nearest point in the base dataset ($\mathcal{B}$) and evenly spread out the projections corresponding to the ECDF interval. We begin with an illustration to explain this in both continuous and discrete cases in Figure 13 and Figure 14 respectively. To define it formally, we first begin with a simpler randomized uniform projection method and then develop the non-randomized version that yields a deterministic low discrepancy sequence, also known as a *centered regular lattice* [Dick and Pillichshammer, 2010, Chapter 1].

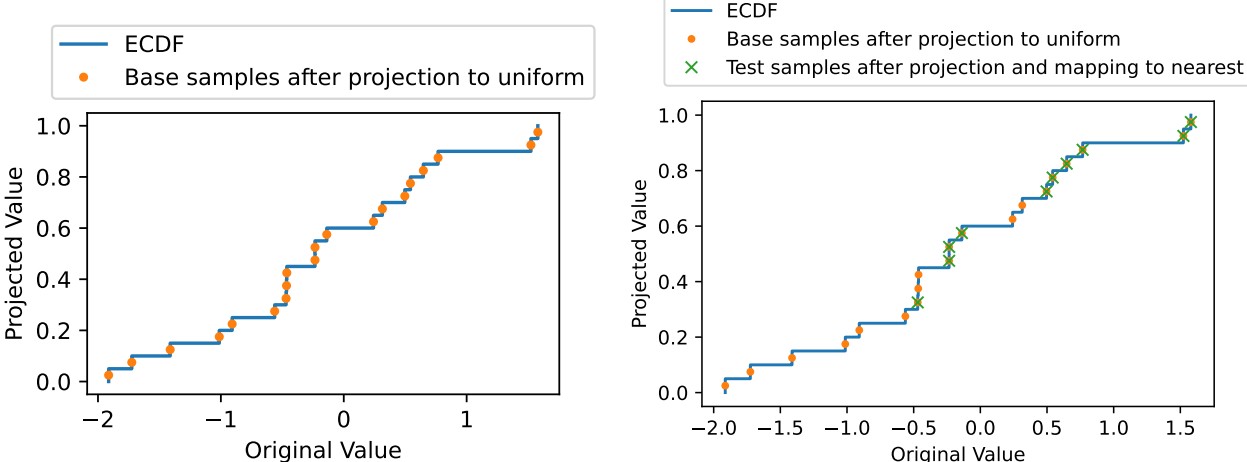

Figure 13: On the left in orange are continuous valued base samples of dataset $\mathcal{B}$ after being projected onto the uniform space using the ECDF. On the right in green are the continuous valued test samples of dataset $\mathcal{A}$ after being projected onto the uniform space using the nearest point in the base dataset $\mathcal{B}$

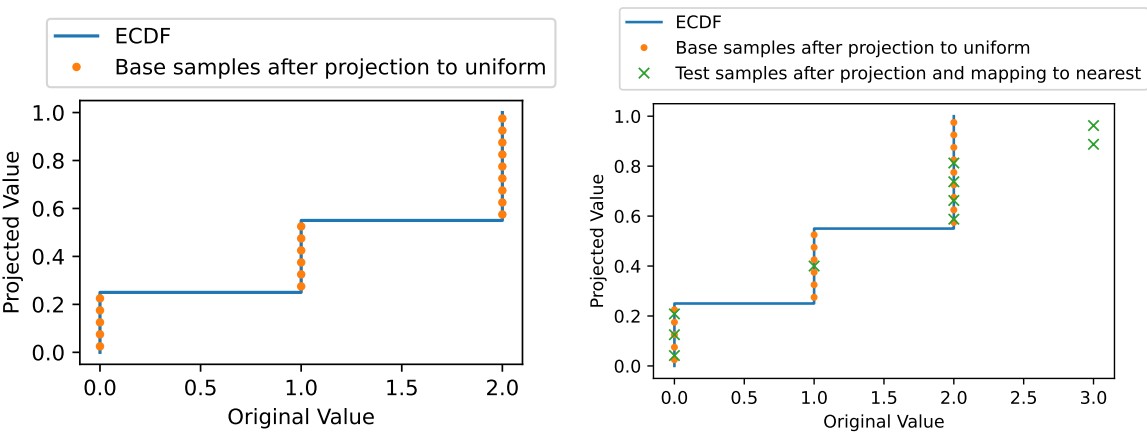

Figure 14: On the left in orange are discrete valued base samples of dataset $\mathcal{B}$ after being projected onto the uniform space using the ECDF and breaking ties between the same points at random. On the right in green are the discrete valued test samples of dataset $\mathcal{A}$ after being projected onto the uniform space using the nearest point in the base dataset $\mathcal{B}$. Note that if the test samples are outside the original data (as the two test points at 3.0), they will be mapped to the nearest base point (2.0 in this figure) and then projected evenly onto the interval spanning the corresponding part of the ECDF.

**Preliminary Notation.** First, let us define the left and right empirical CDFs (the standard ECDF is the right empirical CDF). Informally, these will give us the bottom and top of each "step" in the empirical CDF. Formally, given a 1D dataset $\mathcal{C}$ as:

$$\hat{F}_{\mathcal{C}}^{(-)}(a) = \frac{1}{|\mathcal{C}|} \sum_{c_i \in \mathcal{C}} \mathbf{1}(c_i < a) \tag{8}$$

$$\hat{F}_{\mathcal{C}}^{(+)}(a) = \frac{1}{|\mathcal{C}|} \sum_{c_i \in \mathcal{C}} \mathbf{1}(c_i \le a), \tag{9}$$

where $\mathbf{1}(\cdot)$ is an indicator function that is 1 if the condition is true and 0 otherwise. Note that the only difference between these is that in the left ECDF $\hat{F}_{\mathcal{C}}^{(-)}$, the condition does not include equality.

**Naïve Randomized Projection to Unit Interval.** First, we will consider a randomized projection to the uniform based on the left and right ECDFs defined above. Formally, with two scalar datasets, $\mathcal{A} = \{a_i \in \mathbb{R}\}_{i=1}^{|\mathcal{A}|}$ and a base dataset

$\mathcal{B} = \{b_i \in \mathbb{R}\}_{i=1}^{|\mathcal{B}|}$, we define our function as:

$$F_{\text{rand}}(\mathcal{A}; \mathcal{B}) \triangleq \{(1 - V_i)\hat{F}_{\mathcal{B}}^{(-)}(b^*(a_i)) + V_i \hat{F}_{\mathcal{B}}^{(+)}(b^*(a_i)) : a_i \in \mathcal{A}\}, \tag{10}$$

where $b^*(a_i) := \arg\min_{b \in \mathcal{B}} \|a_i - b\|_2^2$ represents the closest point in $\mathcal{B}$ to $a_i$ and $V_i \sim \text{Uniform}(0, 1)$ is an independent uniform random variable corresponding to each value $a_i \in \mathcal{A}$. This projects each point $a_i$ uniformly over the interval $[\hat{F}_{\mathcal{B}}^{(-)}(b^*(a_i)), \hat{F}_{\mathcal{B}}^{(+)}(b^*(a_i))]$. In the special case where $\mathcal{A} = \mathcal{B}$, i.e., the test points are the same as the base dataset, then this produces a marginally uniform distribution over $[0, 1]$ because each test point is randomly projected on the $1/|\mathcal{B}|$ interval of the ECDF corresponding to itself. Even if there are ties in the dataset (e.g., with a discrete dataset), this will still produce a uniform marginal distribution because the ECDF will jump by the $n_{\text{ties}}/|\mathcal{B}|$, where $n_{\text{ties}}$ is the number of ties for a particular value in $\mathcal{B}$.

**Low Discrepancy Projection to Unit Interval.** The main drawback to the projection method described above is that it is random and may not evenly space out points across the unit interval $[0,1]$, i.e., the sequence may not have low discrepancy compared to evenly spaced points [Dick and Pillichshammer, 2010]. Therefore, we propose a non-randomized version of the above projection method that evenly spaces test points across the corresponding interval instead of choosing a random point in the interval, i.e., we replace $V_i$ with a deterministic function. This form of spacing is known as a regular centered lattice [Dick and Pillichshammer, 2010, Chapter 1]. Formally, we define our non-randomized function as:

$$F(\mathcal{A}; \mathcal{B}) := \{(1 - v(a_i))\hat{F}_{\mathcal{B}}^{(-)}(b^*(a_i)) + v(a_i)\hat{F}_{\mathcal{B}}^{(+)}(b^*(a_i)) : a_i \in \mathcal{A}\}, \tag{11}$$

where $b^*(a_i)$ is the closest point as defined in the randomized version but we replace a random $V_i$ with a deterministic function of the sets defined as follows:

$$v(a_i) := \frac{\text{sorted\_index}(a_i, \mathcal{A}(a_i)) - 0.5}{|\mathcal{A}(a_i)|} \tag{12}$$

$$\mathcal{A}(a_i) := \{a_j \in \mathcal{A} : b^*(a_j) = b^*(a_i)\}. \tag{13}$$

where $\mathcal{A}(a_i)$ represents the equivalence class of all points in $\mathcal{A}$ that have the same closest point in $\mathcal{B}$ and $\text{sorted\_index}(a_i, \mathcal{A}(a_i))$ returns the index (with one-indexing) of the element in the multi-set of $\mathcal{A}(a_i)$ after sorting the elements where ties are broken arbitrarily. Note that $v(a_i) \in [0, 1]$ by construction and it evenly spaces the points in the equivalence class of $\mathcal{A}(a_i)$ over the interval corresponding to the equivalence class. For example, if $a_i = 1$ and $\mathcal{A}(a_i) = \{1, 0.5, 1.5\}$, then $v(a_i) = \frac{\text{index}(a_i, \mathcal{A}(a_i)) - 0.5}{|\mathcal{A}(a_i)|} = \frac{2 - 0.5}{3} = \frac{1.5}{3} = 0.5$, i.e., it would project this point to the center of the interval $[\hat{F}_{\mathcal{B}}^{(-)}(b^*(a_i)), \hat{F}_{\mathcal{B}}^{(+)}(b^*(a_i))]$.

## E.2 COMPUTING THE IMPORTANCE WEIGHTS

We Used Kernel Mean Matching (KMM) [Huang et al., 2006, Yu and Szepesvari, 2012] implemented inde Mathelin et al. [2021] to directly estimate the ratio in Equation 2 using the generated and held-out samples. For this approach we used an RBF kernel and experimented with different values of kernel bandwidth to finally set on using $\frac{10}{\sigma(Z_{i,\ell})}$ where $\sigma(Z_{i,\ell})$ is the standard deviation of the held portion of the data.

## E.3 PROCEDURE FOR GENERATING SAMPLES FROM THE MODEL

Because the samples are re-weighted based on the importance weights, the effective number of samples for statistical tests is less than the number of generated samples. Therefore, we choose to generate the same number of *effective* samples for each method where the number of effective samples is defined as in the main paper but repeated here for clarity:

$$N_{\text{eff}}(\mathcal{G}_{\text{gen},W}^{(\ell,j)}) = \frac{(\sum_{(\bar{G}_i, W(\bar{G}_i)) \in \mathcal{G}_{\text{gen},W}^{(\ell,j)}} W^{(\ell,j)}(\bar{G}_i))^2}{\sum_{(\bar{G}_i, W(\bar{G}_i)) \in \mathcal{G}_{\text{gen},W}^{(\ell,j)}} W^{(\ell,j)}(\bar{G}_i)^2}. \tag{14}$$

Specifically, we set a target threshold of the number of effective samples $t = \min(1000, |\mathcal{G}_{\text{held}}^{(\ell,j)}|)$. We iteratively generated batches of t samples and checked if the concatenated samples reached the number of effective samples, and we stopped generating once it reached the desired value.

# F PROOFS

**Proposition 1.** *For any $\epsilon < 1$ and $\psi \in \{1, 2, \dots\}$ and assuming the splits are equal size in expectation, i.e., $p(S_{i,\ell}) = \frac{1}{k}$, if $p(U_{i,\ell}|S_{i,\ell}) = (1-\epsilon)p_{\text{BetaMix}}(U_{i,\ell}|S_{i,\ell}) + \epsilon p_{\text{Unif}[0,1]}(U_{i,\ell})$, where*

$$p_{\text{BetaMix}}(U_{i,\ell}|S_{i,\ell}=j) = \frac{1}{\psi}\sum_{a=1}^{\psi} p_{\text{Beta}[\alpha_{j,a},\beta_{j,a}]}(U_{i,\ell})$$

*and where $\alpha_{j,a} \triangleq (j-1)\psi + a$ and $\beta_{j,a} \triangleq \psi k + 1 - \alpha_{j,a}$,*

*then $p(U_{i,\ell}) = \text{Uniform}[0,1]$ and the splits will be biased, i.e., $p(S_{i,\ell}|G_i) = p(S_{i,\ell}|U_{i,\ell}) \neq p(S_{i,\ell})$ or equivalently $I(S_{i,\ell}, G_i) > 0$.*

*Proof.* In the proof below we suppress the dependency on $i$ and $\ell$ when needed for simplicity. To prove that $P(U_{i,\ell}) = \text{Uniform}[0,1]$ we first prove that this is true for $p_{\text{BetaMix}}(U_{i,\ell}|S_{i,\ell})$ by marginalizing over the joint distributions of $U_{i,\ell}$ and $S_{i,\ell}$ to get:

$$\sum_{j=1}^{k} p_{\text{BetaMix}}(S=j, U) \tag{15}$$

$$= \sum_{j=1}^{k} p(S=j)p_{\text{BetaMix}}(U|S=j) \tag{16}$$

$$= \sum_{j=1}^{k} P(S=j)\frac{1}{\psi}\sum_{a=1}^{\psi} p_{\text{Beta}[\alpha_{j,a},\beta_{j,a}]}(U) \tag{17}$$

$$= \frac{1}{\psi K}\sum_{j=1}^{k}\sum_{a=1}^{\psi} p_{\text{Beta}[\alpha_{j,a},\beta_{j,a}]}(U) \tag{18}$$

$$= \frac{1}{\psi K}\sum_{j=1}^{k}\sum_{a=1}^{\psi} p_{\text{Beta}[(j-1)\psi+a, k\psi+1-((j-1)\psi+a)]}(U) \tag{19}$$

Let $n = k\psi$, $r = (j-1)\psi + a$ then we can rewrite the above as:

$$= \frac{1}{n}\sum_{r=1}^{n} p_{\text{Beta}[r, n+1-r]}(U) \tag{20}$$

$$= p_{\text{Unif}[0,1]}(U) \tag{21}$$

Equation 20 is similar to Segers et al. [2017] results in section 2.1 if we substitute $d = 1$ and $n = k\psi$ in their results, and use that to arrive to the last equality leading to Equation 21.

Next, we show that this holds true for $p(U|S)$ as follows:

$$\sum_{j=1}^{k} p(S=j, U) \tag{22}$$

$$= \sum_{j=1}^{k} p(S=j)p(U|S=j) \tag{23}$$

$$= \sum_{j=1}^{k} p(S=j)[(1-\epsilon)p_{\text{BetaMix}}(U|S) + \epsilon p_{\text{Unif}[0,1]}(U)] \tag{24}$$

$$= \epsilon p_{\text{Unif}[0,1]}(U) + (1-\epsilon)\sum_{j=1}^{k} p(S=j)p_{\text{BetaMix}}(U|S=j) \tag{25}$$

$$= p_{\text{Unif}[0,1]}(U). \tag{26}$$

Invoking the previous results, we see that the above is also uniform.

To prove that the splits will be biased, we will use mutual information as follows: Let $p(U, S)$ denote the joint distribution of $p(U_{i,\ell}, S_{i,\ell})$. The mutual information for $\epsilon < 1$ can be written as:

$$I(U, S) \equiv \mathrm{KL}(p(U, S), p(U)p(S)) \tag{27}$$

$$= \mathbb{E}_{p(S)}[\mathbb{E}_{p(U|S)}[\log \frac{p(U, S)}{p(U)p(S)}]] \tag{28}$$

$$= \mathbb{E}_{p(S)}[\mathbb{E}_{p(U|S)}[\log \frac{p(U|S)p(S)}{p(U)p(S)}]] \tag{29}$$

$$= \mathbb{E}_{p(S)}[\mathbb{E}_{p(U|S)}[\log \frac{p(U|S)}{p(U)}]] \tag{30}$$

$$= \mathbb{E}_{p(S)}[\mathrm{KL}(p(U|S), p(U))]] \tag{31}$$

$$> 0, \tag{32}$$

where the last inequality is because $p(U)$ is a uniform distribution but $p(U|S = j)$ is not uniform whenever $\epsilon < 1$, i.e., if $\epsilon < 1$, then $p(U|S = j) \neq p(U), \forall j$, and thus the KL must be positive for all terms in the expectation.

$\square$

**Remark on Mutual Information Between Splits** We also briefly discuss additional design points of the mutual information between splits and graphs. If $\epsilon = 0$ and $\psi = 1$, since $p(U) = p_{\mathrm{Unif}[0,1]}$, we also know that the KL terms are equal to negative differential entropy of the Beta distributions which is known in closed form, i.e.,

$$\mathrm{KL}(p_{\mathrm{Beta}[\alpha,\beta]}, p_{\mathrm{Unif}[0,1]}) \equiv -H(p_{\mathrm{Beta}[\alpha,\beta]})$$
$$= -[\log \mathrm{B}(\alpha, \beta) - (\alpha - 1)\gamma(\alpha) - (\beta - 1)\gamma(\beta) + (\alpha + \beta - 2)\gamma(\alpha + \beta)], \tag{33}$$

where $\mathrm{B}(\cdot, \cdot)$ denotes the the Beta function and $\gamma(\cdot)$ denotes the digamma function.

Furthermore, if $0 \leq \epsilon \leq 1$ and we consider the term $\mathrm{KL}((1 - \epsilon)p_{\mathrm{Beta}[\alpha,\beta]} + \epsilon p_{\mathrm{Unif}[0,1]}, p_{\mathrm{Unif}[0,1]})$ which we will refer to as $\mathrm{KL}_\epsilon$, we know that at $\epsilon = 1$ the term becomes $\mathrm{KL}(p_{\mathrm{Unif}[0,1]}, p_{\mathrm{Unif}[0,1]}) = 0$, and on the other extreme at $\epsilon = 0$, it becomes $\mathrm{KL}(p_{\mathrm{Beta}[\alpha,\beta]}, p_{\mathrm{Unif}[0,1]})$ which is known in closed form by the result above. Thus for any $0 < \epsilon < 1$, we are guaranteed to have: $0 < \mathrm{KL}_\epsilon < -H(p_{\mathrm{Beta}[\alpha,\beta]})$.

**Theorem 1** ($\phi_{\mathrm{KS}}(\mathcal{G}_{\mathrm{held},\mathbf{1}}^{(\ell,j)}, \mathcal{G}_{\mathrm{gen},W}^{(\ell,j)}; h_{\ell'})$ consistent)**.** *Using VV for generating data splits and corresponding datasets* $\mathcal{G}_{train}^{(\ell,j)}$, $\mathcal{G}_{held}^{(\ell,j)}$ *and using an implicit generator* $\Omega$ *trained on* $\mathcal{G}_{train}^{(\ell,j)}$ *to generate data* $\mathcal{G}_{gen}^{(\ell,j)}$.

*Then, if* $\mathcal{G}_{gen}^{(\ell,j)}$ *is generated with the same distribution as* $\mathcal{G}_{held}^{(\ell,j)}$, *for any* $\epsilon \in [0, 1]$,

$$P(\phi_{KS}(\mathcal{G}_{held,\mathbf{1}}^{(\ell,j)}, \mathcal{G}_{gen,W}^{(\ell,j)}; h_{\ell'}) > \epsilon) \leq$$
$$4 \exp \left( -2 \min(|\mathcal{G}_{gen}^{(\ell,j)}|, |\mathcal{G}_{held}^{(\ell,j)}|) \left( \frac{\epsilon}{2} \right)^2 \right), \tag{7}$$

*Proof.* Let $F$ be the true data CDF we would like to generate, i.e., the true CDF of the graph property we are testing. Let $\hat{F}_{n_i}^{(i)}$ be the empirical CDF obtained from the implicit model after $n_i$ samples.[3] Let $\hat{F}_{n_h}^{(h)}$ be the empirical heldout CDF from $n_h$ heldout samples. Finally, let $\hat{F}_{n_m}^{(m)}$ be the empirical memorization CDF with $n_m$ memorization samples (we will use $\hat{F}_{n_m}^{(m)}$ rather than $F_m$ in the theorem statement). By the Dvoretzky-Kiefer-Wolfowitz inequality with Massart's universal constant [Massart, 1990] we have that for an arbitrary empirical distribution $\hat{H}_n$ with $n$ samples and its true distribution $H$,

$$P(\phi_{\mathrm{KS}}(\hat{H}_n, H) > d) \leq 2 \exp(-2nd^2), \forall d > 0.$$

Note that $\phi_{\mathrm{KS}}$ satisfies the triangle inequality, and hence,

$$\phi_{\mathrm{KS}}(\hat{F}_{n_i}^{(i)}, \hat{F}_{n_h}^{(h)}) \leq \phi_{\mathrm{KS}}(\hat{F}_{n_i}^{(i)}, F) + \phi_{\mathrm{KS}}(F, \hat{F}_{n_h}^{(h)}). \tag{34}$$

By the total law of probabilities,

$$P(\phi_{\mathrm{KS}}(\hat{F}_{n_i}^{(i)}, F) < d) \geq 1 - 2 \exp(-2n_i d^2)$$

and

$$P(\phi_{\mathrm{KS}}(F, \hat{F}_{n_h}^{(h)}) < d) \geq 1 - 2 \exp(-2n_h d^2),$$

---

[3]While we prove the theorem statement with respect to an unweighted empirical CDF for the generated samples, we expect a similar result to hold for a weighted empirical CDF where we use the number of effective samples as defined in the main paper in place of $n_i$.

which together with Equation (34) yields

$$P(\phi_{\text{KS}}(\hat{F}_{n_i}^{(i)}, \hat{F}_{n_h}^{(h)}) < d) \geq (1 - 2\exp(-2n_i d^2))(1 - 2\exp(-2n_h d^2))$$

$$\geq 1 - 2\exp(-2n_i d^2) - 2\exp(-2n_h d^2) + 4\exp(-2(n_h + n_i)d^2).$$

By the total law of probabilities, we obtain

$$P(\phi_{\text{KS}}(\hat{F}_{n_i}^{(i)}, \hat{F}_{n_h}^{(h)}) > d) \leq 2\exp(-2n_i d^2) + 2\exp(-2n_h d^2) - 4\exp(-2(n_h + n_i)d^2)$$

$$\leq 2\exp(-2\min(n_i, n_h)d^2) + 2\exp(-2\min(n_i, n_h)d^2)$$

$$\leq 4\exp(-2\min(n_i, n_h)d^2)$$

$$\leq 4\exp\left(-2\min(n_i, n_h)\left(\frac{d}{2}\right)^2\right)$$

Now, set $\epsilon \in [0, 1]$, then we have: $P(\phi_{\text{KS}}(\mathcal{G}_{\text{held},\mathbf{1}}^{(\ell,j)}, \mathcal{G}_{\text{gen},W}^{(\ell,j)}; h_{\ell'}) > \epsilon) \leq 4\exp\left(-2\min(|\mathcal{G}_{\text{gen}}^{(\ell,j)}|, |\mathcal{G}_{\text{held}}^{(\ell,j)}|)\left(\frac{\epsilon}{2}\right)^2\right)$

□

# G  ADDITIONAL EXPERIMENTAL DETAILS AND RESULTS

In this section, we provide additional experimental details concerning the datasets, training setup, as well as additional results and figures.

## G.1  MORE INFORMATION ABOUT THE DATASETS

**Community datasets:**  We used two variations of this dataset (referred to as Comm20 in section 4.1 and as Comm in section 4.2.2) in our experiments. Both variations share the following, they are synthetic graphs made up by exactly two equally sized communities, with number of nodes ranging from 12 to 20. Each of the communities are generated by the random graph generator model, E-R model [Erdős and Rényi, 1960] with $p = 0.7$, and the number of edges added between the two communities is with probability of $0.05$ times the number of nodes in the graph. The variation used in section 4.1 was generated using the parameters above using the code provided in the git repository of the official GDSS model implementation [4] after adjusting the seeds in such a way to avoid exactly repeating the same graph, and we generated 500 samples.

The variation used by the experiments in section 4.2.2 was the original dataset available at the same GDSS repository [5] which contains 100 graphs. After close inspection of this dataset, we noted that it has multiple repeated graphs, due to running the code with similar seeds more than once. In some sense these replications can be considered a form of leakage when splitting the data, however with our approach since we are doing vertical splitting, identical samples are more likely to belong to only v-train or only v-test. We leave it as a future work to thoroughly understand the effects of such repetitions.

**Qm9:**  This dataset [6] contains 134k drug-like molecules which made up of at most 9 heavy (non Hydrogen) atoms: Carbon (C), Oxygen (O), Nitrogen (N), and Flourine (F), and Hydrogen (H) bonding.

The dataset we ended up using after some preprocessing and filtering had a total number of 130,831 samples. We arrived at this number by prepossessing the dataset by removing the hydrogen atoms from all the molecules, and removing all the molecules where any of the 5 properties cannot be calculated by rdkit package. The 5 chosen properties for this dataset are average degree ($\phi_{ks}^{\text{D}}$), molecular weight ($\phi_{ks}^{\text{Mlwt}}$), Topological Polar Surface Area (TPSA) ($\phi_{ks}^{\text{TPSA}}$) [Prasanna and Doerksen, 2009], ring counts ($\phi_{ks}^{\text{RC}}$), and the logarithm of the partition coefficient (logp) ($\phi_{ks}^{\text{LogP}}$).

## G.2  TRAINING DETAILS FOR EACH GRAPH GENERATIVE MODEL

We chose the models: **DiGress** [Vignac et al., 2022], **GDSS** [Jo et al., 2022] and **GGAN** [Krawczuk et al., 2021] to evaluate.

---

[4] https://github.com/harryjo97/GDSS/blob/master/data/data_generators.py
[5] https://github.com/harryjo97/GDSS/blob/master/data/community_small.pkl
[6] https://deepchemdata.s3-us-west-1.amazonaws.com/datasets/molnet_publish/qm9.zip

**DiGress Training.** for the experiments of section 4.2.2 we trained DiGress on the Comm dataset, we set the number of epochs to be 100,000 with a batch size of 256, a learning rate of 0.0002, and an AdamW optimizer. The model parameters used for training were T = 500 diffusion steps, with a cosine noise schedule and 8 layers. Those were the default parameters provided in the config files in the official code repository[7].

For training DiGress with Qm9 dataset, we set the number of epochs to be 1000 with a batch size of 512, a learning rate of 0.0002, a weight decay of $10^{-12}$ and an AdamW optimizer. The model parameters used for training were T = 500 diffusion steps, with a cosine noise schedule and 9 layers.

For the experiments of section 4.2.1, we used the same settings as mentioned above for the respective dataset but trained with early stopping with a patience of 20 and monitoring the Negative log-likelihood (NLL) of the validation data (v-val).

**GDSS Training** The code used for GDSS is adapted from their official repository [8]. The hyperparameters applied to train the Community and Qm9 datasets are taken from the config files provided by the the author [Jo et al., 2022]. For the Community datasets[9], the number of epochs is 5000 with a batch size of 128, a learning rate of 0.01, with an Adam optimizer and Exponential Moving Average (EMA). For Qm9 dataset[10], the number of epochs is 300 with a batch size of 1024, a learning rate of 0.005, with an Adam optimizer and Exponential Moving Average (EMA).

For the experiments in section 4.2.1, we incorporated an early stopping criterion. Specifically, training was terminated if the difference between the MMD loss (estimated partial scores, equation 5 in Jo et al. [2022]) for the training and validation sets did not decrease below $1e - 10$ for 5 consecutive epochs. This prevents overfitting and saves computational resources.

**GGAN Training** GGAN code is taken from SPECTRE [Martinkus et al., 2022] repository [11] with --use_fixed_emb argument while training. Only Comm dataset is used with this model. The hyper-parameters to train the model are also taken from the suggested commands for Comm in the mentioned repository: the number of epochs is up to 12000 with a batch size of 10, a learning rate of 0.0001, with an Adam optimizer (both discriminator and generator).

## G.3 CHOOSING THE SPLIT FEATURE

In section 4.1 we chose a single split property out of 3 possible properties, and in section 4.2.1 we chose one out of 5 possible properties. Intuitively, a split property which is completely independent of the other test properties would not cause any (indirect) shift of the other properties. Thus, the marginal distributions of each property would be equal for every split. This would not enable good evaluation of the generalization performance on the thin support since a model could just copy the marginal distributions of the properties from the training split. Therefore, we choose to find a split property that has high dependence with all other split properties. Concretely, in both cases we calculated the spearman correlation between all the features against each other and computed the average absolute correlation that one feature has with the rest, then choose the feature that has the highest average correlation to be the split feature. As we have mentioned in the discussion, this is still an area of exploration, but the motivation behind this approach was that choosing a split feature that is somewhat correlated with the others will cause the distribution of the other features to change accordingly which is the desirable effect in our case.

## G.4 ECDF PLOTS FOR MODEL COMPARISON

In this section we present a list of ECDF plots for some splits that we inspected for further analysis.

In the main paper we mentioned two conjectures and that we arrived to them by inspecting some ECDF plots, we point the reader to those ECDF plots below. First, the conjecture that GGAN generally matches the modes of the distribution while DiGress and GDSS tend to be more smooth, this behaviour is emphasized in Figure 15, Figure 18, Figure 16, Figure 17.

Second, the conjecture that GDSS and DiGress are not able to concentrate the generation near the middle of the distribution for test property average shortest path length, which can be seen in Figure 19, and Figure 20.

---

[7]https://github.com/cvignac/DiGress
[8]https://github.com/harryjo97/GDSS/tree/master
[9]https://github.com/harryjo97/GDSS/blob/master/config/community_small.yaml
[10]https://github.com/harryjo97/GDSS/blob/master/config/qm9.yaml
[11]https://github.com/KarolisMart/SPECTRE/tree/main

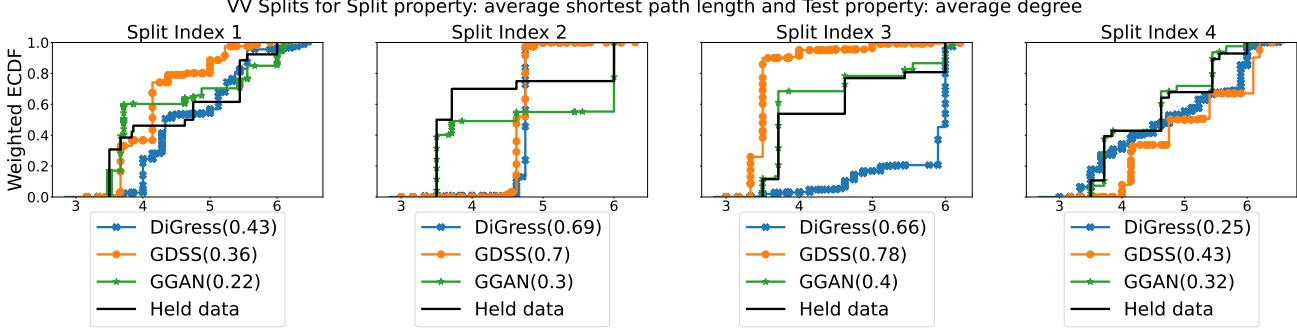

Figure 15: Weighted ECDF for DiGress, GDSS and GGAN as compared to held out data for split property average average shortest path length ($\ell = 3$) when test property is average degree ($\ell' = 1$). The legend reads modelName($\phi_{ks}$) value

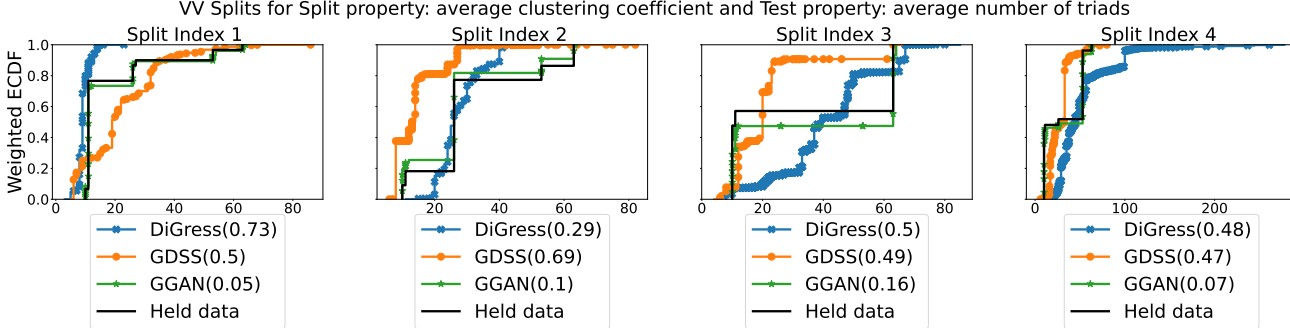

Figure 16: Weighted ECDF for DiGress, GDSS and GGAN as compared to held out data for split property average clustering coefficient ($\ell = 4$) when test property is average number of triads ($\ell' = 2$). The legend reads modelName($\phi_{ks}$) value

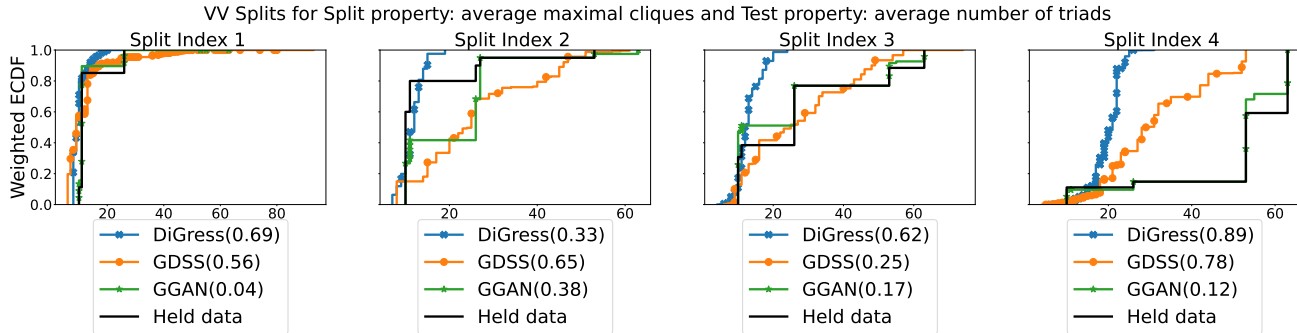

Figure 17: Weighted ECDF for DiGress, GDSS and GGAN as compared to held out data for split property average maximal cliques ($\ell = 5$) when test property is average number of triads ($\ell' = 2$). The legend reads modelName($\phi_{ks}$) value

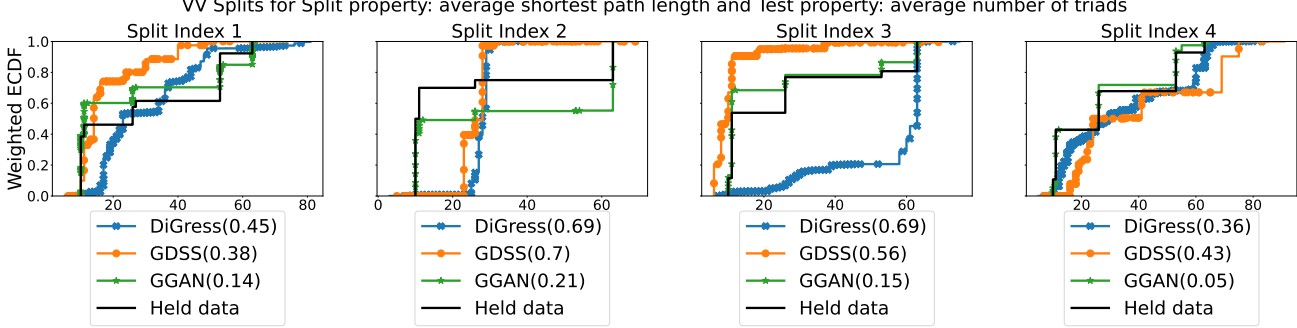

Figure 18: Weighted ECDF for DiGress, GDSS and GGAN as compared to held out data for split property average shortest path length ($\ell = 3$) when test property is average number of triads ($\ell' = 2$). The legend reads modelName($\phi_{ks}$) value

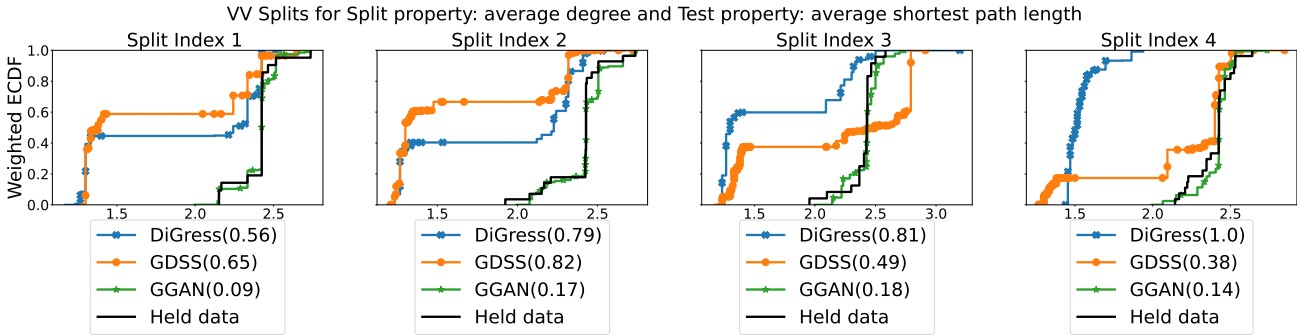

Figure 19: Weighted ECDF for DiGress, GDSS and GGAN as compared to held out data for split property average degree ($\ell = 1$) when test property is average shortest path length ($\ell' = 3$). The legend reads modelName($\phi_{ks}$) value

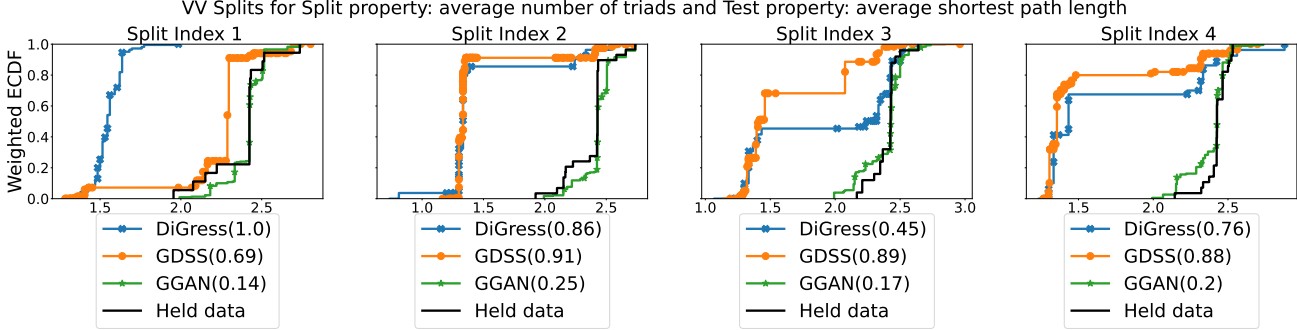

Figure 20: Weighted ECDF for DiGress, GDSS and GGAN as compared to held out data for split property average number of triads ($\ell = 2$) when test property is average shortest path length ($\ell' = 3$). The legend reads modelName($\phi_{ks}$) value

## G.5 EXAMPLES OF GENERATED MOLECULES

From the top 100 molecules generated from DiGress and GDSS we filter for validity and novelty, and we visualize the top 4 weighted molecules in different scenarios.

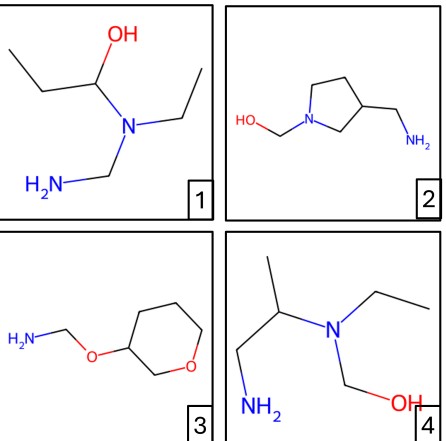

(a) Top 4 valid and novel molecules sampled from DiGress when evaluated against v-val

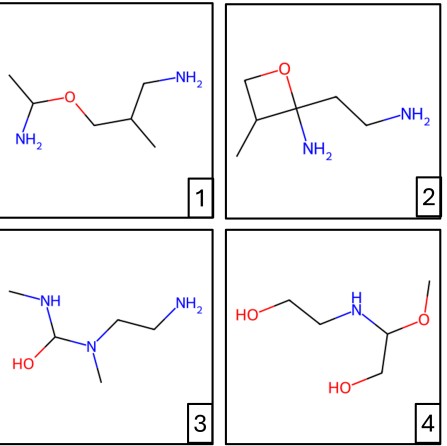

(b) Top 4 valid and novel molecules sampled from DiGress when evaluated against v-test

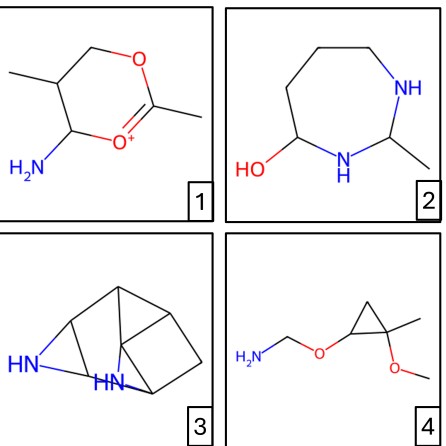

(c) Top 4 valid and novel molecules sampled from GDSS when evaluated against v-val

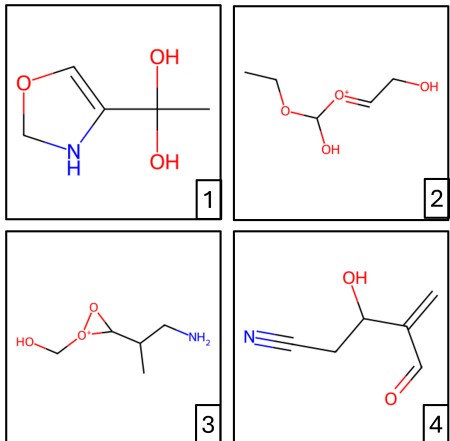

(d) Top 4 valid and novel molecules sampled from GDSS when evaluated against v-test

Figure 21: Visualization of the top 4 highest weighted molecules (after filtering for novelty and validity) that are sampled from DiGress and GDSS and assigned weights by our approach when testing against the held portion being either v-val or v-test. These are the same molecules in Figure 6 that were plotted with respect to the TPSA distribution.