# OpenReview forum: "Vertical Validation: Evaluating Implicit Generative Models for Graphs on Thin Support Regions"
_auai.org/UAI/2024/Conference — UAI 2024 poster_

### Official Review · Reviewer_3ddL · 2024-02-27

**Q2-1 Originality-Novelty:** 3
**Q2-2 Correctness-Technical Quality:** 3
**Q2-5 Clarity Of Writing:** 4

**Q1 Summary And Contributions:**

A new framework named Vertical Validation (VV) for biased splitting and reweighting is proposed to evaluate the generalizability of implicit graph generative models. The results of comparative experiments and theoretical proofs verify the effectiveness of VV.

**Q2-3 Extent To Which Claims Are Supported By Evidence:**

4: Excellent: all claims are supported by very convincing evidence (in the form of comprehensive experimental evaluation, rigorous mathematical proofs, detailed (pseudo-)code, precise references, well-motivated and realistic assumptions) and the authors deliver what they promise.

**Q2-4 Reproducibility:**

3: Good: key resources (e.g. proofs, code, data) are available and key details (e.g. proofs, experimental setup) are sufficiently well-described for competent researchers to confidently reproduce the main results.

**Q3 Main Strengths:**

The paper is well-organized and clearly written.

This paper is rich in experimental and theoretical proof.

The research content of this paper is relatively novel.

**Q4 Main Weakness:**

The proposed evaluation method is not applied to the latest generation model.

**Q5 Detailed Comments To The Authors:**

The proposed evaluation method is only applied to representative generative models. Can the authors apply this evaluation method to the latest (2023, 2024) generation models?Please show some experimental results if possible.

**Q9 Complying With Reviewing Instructions:**

Yes

---

> ### Author Rebuttal · Authors · 2024-04-07
>
> We thank the reviewer on their comments and acknowledging the organization and novelty of our approach. Concerning comparing the most recent graph generative models; due to the short time available for the rebuttal period, the time consuming nature of setting up and training a new model, and the fact that graph generative models are much less standardized and less developed than image generative models with varying experimental setups and diverse code bases, we are unfortunately unable to provide results on additional models at the moment.

---

### Official Review · Reviewer_H42J · 2024-03-21

**Q2-1 Originality-Novelty:** 3
**Q2-2 Correctness-Technical Quality:** 3
**Q2-5 Clarity Of Writing:** 3

**Q10 Ethical Concerns:**

No.

**Q1 Summary And Contributions:**

The article initiates a discussion on the significance and challenges involved in evaluating implicit graph generative models, particularly for applications such as molecule design. It highlights the limitations of conventional evaluation techniques that primarily focus on statistical metrics like mean and variance within densely supported regions, which do not adequately assess the models' capability to generate new and potentially useful graphs outside these well-explored areas. To bridge this evaluation gap, the authors propose a novel method named Vertical Validation (VV). This method introduces the concept of thin support regions during the train-test splitting phase and incorporates a reweighting mechanism for a more refined comparison of generated samples with the held-out test data. Such an approach enables a more detailed evaluation of a model’s proficiency in generating innovative structures.

**Q2-3 Extent To Which Claims Are Supported By Evidence:**

3: Good: the main claims are supported by convincing evidence (in the form of adequate experimental evaluation, proofs, (pseudo-)code, references, assumptions).

**Q2-4 Reproducibility:**

3: Good: key resources (e.g. proofs, code, data) are available and key details (e.g. proofs, experimental setup) are sufficiently well-described for competent researchers to confidently reproduce the main results.

**Q3 Main Strengths:**

This article is clearly written and the Vertical Validation mechanism is carefully designed.

**Q4 Main Weakness:**

Concerns:
If the objective is to generate molecules with two optimized properties, and these two molecular properties are intrinsically correlated, possibly by an unknown complex nonlinear mapping, is the proposed splitting approach still viable?

**Q5 Detailed Comments To The Authors:**

See the above comments.

**Q9 Complying With Reviewing Instructions:**

Yes

---

> ### Author Rebuttal · Authors · 2024-04-07
>
> We thank the reviewer for their insightful question!
> To answer your question on whether our approach is still viable if the properties are intrinsically correlated by an unknown complex linear mapping, in short, yes, the approach would still be viable.
> In fact, our evaluation method is most effective when there is high dependency between graph properties because then our systematic shift along the split property will (implicitly) create marginal distribution shifts along the test properties that will be detectable via the KS statistic.
> Please see the example in Figure 1 for data that lies on a circle. In this case, the two properties are intrinsically correlated via a complex non-linear mapping. The shifted splits along the x-axis as in Figure 1, yield different test marginal distributions along the y-axis. Therefore, indeed, highly dependent properties could be helpful for our evaluation method and would be perfectly viable. We will be sure to add this discussion to the final paper.

---

### Official Review · Reviewer_TMBC · 2024-03-25

**Q2-1 Originality-Novelty:** 2
**Q2-2 Correctness-Technical Quality:** 3
**Q2-5 Clarity Of Writing:** 4

**Q1 Summary And Contributions:**

The authors propose a novel evaluation method for graph generative models. The problem in question is the lack of metrics assessing generative model quality in the regions with small number of data points, additionally motivated by the fact that exploration of these regions is often the end goal of using a generative model. The core contribution of the paper is a technique of artificial thinning out of the complementary parts of train and test datasets, split according to a certain graph property.

**Q2-3 Extent To Which Claims Are Supported By Evidence:**

3: Good: the main claims are supported by convincing evidence (in the form of adequate experimental evaluation, proofs, (pseudo-)code, references, assumptions).

**Q2-4 Reproducibility:**

3: Good: key resources (e.g. proofs, code, data) are available and key details (e.g. proofs, experimental setup) are sufficiently well-described for competent researchers to confidently reproduce the main results.

**Q3 Main Strengths:**

The paper is technically solid, experimental part is thought through and thorough, proposed approach is theoretically well supported. The problem discussed is topical, and the presented solution is a good practical approach to address it.

**Q4 Main Weakness:**

The level of novelty of the paper is unclear. The technique is quite general, and there is no clear motivation why it is graph-specific. Actual examples of the graphs generated are missing, making it impossible to relate to the presented results, especially in the context of motivation. The proposed method requires additional training, which is not common for evaluation methods and should be clearly communicated early in the paper.

**Q5 Detailed Comments To The Authors:**

Another round of proof-reading would be beneficial (e.g. lowercase ks, diacritical signs).
"Thick" and "thin" terms are a bit misleading from a mathematical point of view as they suggest the difference in dimensionality of the data manifold. Additionally, these terms should be properly defined in the first paragraphs.
The motivation part is somewhat vague, because the search for new molecules actually often targets "thick" regions of the data for practical reasons, and it is essentially a fast approximation of a combinatorial search on a set of pre-defined molecular fragments. When talking about the discovery of molecules with drastically different structure, this search is inherently out of distribution and not directly related to the proposed work. This motivation also contrasts with the experiments a little - discussion for Qm9 is in the appendix and does not feature molecules.
As mentioned above, actual examples of generated graphs would improve the readability of the discussion part.
The overall technique appears to be more suitable for dropout-like training, wrapping it as evaluation method is somewhat awkward. Maybe some discussion around it would make the paper more organic.
A provided code could really help with trying out this method in practice.

**Q9 Complying With Reviewing Instructions:**

Yes

---

> ### Author Rebuttal · Authors · 2024-04-07
>
> We thank the reviewer for their feedback and time! Below we attempt to address your concerns:
> >Another round of proof-reading would be beneficial
>
> We agree and we are in the process of performing another round of proof reading and fixing typos.
> >"Thick" and "thin" terms are a bit misleading
>
> Thank you for pointing that out. We will work to more carefully define and explain those terms early on.
> > ... the search for new molecules actually often targets "thick" regions
>
> Thank you for your insight! We agree on your point that novel molecule generation tasks may focus on graphs with similar structures, i.e., parts of the distributions where the marginal graph properties are similar to known molecules. However, we want to point out that "thin" support regions in the joint distribution could be hidden in the "thick" support of marginal distributions, especially when considering a distribution in high dimensions. For example, consider samples on a 3D sphere. When projected onto almost any direction, it will look like the support is dense near zero. However, the distribution has no support at or near the all zero vector. Thus, we hypothesize that in high dimensional spaces, there are many thin support regions that are hidden. When we systematically create thin support regions using our approach, the goal is to measure the model's ability to generalize to thin support in general (including hidden thin support). Thus while aiming to generate for thin support can be unrealistic for some properties, we test the ability of the model to generate in those regions as this will reflect its ability to generate in hidden thin support.
>
> Concerning the motivation, we picked the molecule generation example in the introduction as it is a commonly used graph generation task that will allow us to better explain the general goals of our approach. However, the scope and main goal of our approach expands beyond that to include its use in evaluation (or model selection) with respect to generalization on thin support for any graph generating generative model regardless of the type of graphs generated. The motivation for being graph specific is due to the lack of standardization of the metrics used in the graph generation domain, and also because for graphs we can exactly compute their properties which is needed for our approach to work.
> > actual examples of generated graphs would improve the readability of the discussion part
>
> We attempt to do so [here](https://anonymous.4open.science/r/Additions-850B/_UAI_2024__new_addition.pdf) and we agree on the importance of providing some visual examples in the final version of the paper.
> We briefly described those experiments in an answer to the first reviewer—and in the link in more details —that we copied below for completeness. [Copied answer begins here:] we attempted to visualize the top four highest weighted novel molecules generated by both GDSS and DiGress from two training scenarios created using VV, and add plots to show where they lie with respect to the property distribution. We also show that among the top 100 highest weighted generated molecules, several were found to exactly match those in the held out data, which suggests that these molecules are viable novel molecules.
>
> > The overall technique appears to be more suitable for dropout-like training, wrapping it as evaluation method is somewhat awkward.
>
> Thanks for bringing this to our attention, however we admit that we are unclear about what "dropout-like training" refers to? we would appreciate it if you could clarify that. We will briefly attempt to provide a general overview of our approach and how it relates to evaluation. Step 1 of our approach creates biased train-held splits, where a portion of the samples is assigned to each split. We would say that this step is similar to cross validation (but different in the sense that CV is assigning samples to splits uniformly at random but we are assigning samples to splits based on some calculated property of the sample). Then step 2 introduces a way that systematically evaluates the model using its produced samples in a way that accounts for the bias that we intentionally introduced in step 1. We hope this answer provides some clarification and we are happy to discuss it more.
>
> >  A provided code could really help with trying out this method in practice
>
> The code for our approach is provided in the uploaded supplementary material in the file "propertysplit2.py". We also provide 2 notebooks that were used to generate the results in section 4.1. Those notebooks also serve as examples for how to use the code in "propertysplit2.py". The graph generative models we used are all open sourced and their code links are provided in our paper, we did however modify some aspects of their data loading capabilities to be able to easily utilize our computed splits. We will be sure to publish all our modifications to a public repository should our paper be accepted.

---

### Official Review · Reviewer_uKVR · 2024-03-28

**Q2-1 Originality-Novelty:** 3
**Q2-2 Correctness-Technical Quality:** 3
**Q2-5 Clarity Of Writing:** 2

**Q1 Summary And Contributions:**

This paper proposes a new method, VV (vertical validation) for evaluating a graph generative model's ability to generate novel graphs on thin support regions. Existing evaluation methods for graph generative models are mean-based and focus on regions with thick support, thus may fail to correctly evaluate the performance of a graph generative model on thin support regions. VV uses a "vertical" train-test data splitting approach to create thin support in the training set while the test set has ground truth samples from these thin support region for evaluating a model's ability to generate graphs in the thin support regions. VV also uses a reweighting strategy for an unbiased evaluation using the biasedly split train-test data. Experiments are conducted with two datasets on three generative models to show the effectiveness of VV.

**Q2-3 Extent To Which Claims Are Supported By Evidence:**

2: Fair: the main claims are somewhat supported by evidence (but the experimental evaluation may be weak, or does not match entirely with the claims, important baselines may be missing, proofs contain important ideas but lack rigor, algorithmic details are only discussed superficially, references are imprecise, assumptions are not sufficiently motivated or explicated, etc.).

**Q2-4 Reproducibility:**

3: Good: key resources (e.g. proofs, code, data) are available and key details (e.g. proofs, experimental setup) are sufficiently well-described for competent researchers to confidently reproduce the main results.

**Q3 Main Strengths:**

1. The problem addressed is meaningful and useful.
2. The proposed "vertical" split seems novel and interesting.

**Q4 Main Weakness:**

1. The clarity of the presentation of some technical details should be improved, although overall the quality of presentation is good.
2. There does not seem to be a theoretical guarantee on obtaining a desired biased split by VV.
3. The discussion of the originality of the proposed idea can be enhanced.
4. The experiments can be enhanced too.

**Q5 Detailed Comments To The Authors:**

More details about the above listed weakness:
1. Presentation: It's not very obvious to me how the two desired qualities of the splitting method could lead to biased splits. Could you elaborate on this?
2. Is there a theoretical guarantee that the splits created using the method presented in Section 3.1 will be biased as wished?
3. In the image generation domain, are there any evaluation methods/metrics which have considered images in thin support regions? Would it be possible to adapt those methods/metrics to graph generative model evaluation?
4. The paper is motivated by using graph generative models to design or discover new molecules for medicine or material design. It would be useful to demonstrate in the experiments by showing some examples of new molecules generated or discovered by graph generative models which may have potential for medicine or material design.

**Q9 Complying With Reviewing Instructions:**

Yes

---

> ### Author Rebuttal · Authors · 2024-04-07
>
> Thank you for your feedback and time! We attempt to address your comments below:
> > It's not very obvious to me how the two desired qualities of the splitting method could lead to biased splits.
>
> Creating biased splits means that the split variable depends on the graph, i.e., $P(S_{i,\ell} | G_i) \neq P(S_{i,\ell})$. Many distributions of $P(S_{i,\ell}|G_i)$ could give biased splits, but we wanted both a generic and balanced splitting method. Thus, the two qualities/properties specify the constraints rather than the objective of our splitting method. We apologize for this confusion and will clarify it in the paper. Concretely, the first quality constrains the space of distributions to those that only depend on $U_{i,\ell}$, which is a function of $G_i$. Since $U_{i,\ell}$ encodes the normalized rank information of the split property, it can be applied to any property distribution. The second quality ensures that the splits are balanced. Using Bayes rule, we show that because $U_{i,\ell}$ will have a uniform distribution, we can simplify the constraint to finding a component distribution whose mixture is a uniform distribution and whose weights are equal to $p(S_{i,\ell})$. This leads us to using specific Beta distributions as components as their mixture is equal to a uniform distribution (and thus lead us to balanced splits). Assuming that the mixture components are not equal, it is simple to show that $p(S_{i,\ell} | G_i) =  p(S_{i,\ell} | U_{i,\ell}) = \frac{p(S_{i,\ell})p(U_{i,\ell}|S_{i,\ell})}{p(U_{i,\ell})}  \neq p(S_{i,\ell})$ as $p(U_{i,\ell}|S_{i,\ell}) \neq p(U_{i,\ell})$ except in the special case where $p(U_{i,\ell}|S_{i,\ell})$ is uniform given any split (which correspond to standard random splitting). Given all this and our final choice for $p(U_{i,\ell}|S_{i,\ell}) = (1-\epsilon) p(U_{i,\ell}|S_{i,\ell}) + \epsilon . p_{\mathrm{Unif}[0,1]}(U_{i,\ell})$ our splitting distributions will be biased as long as $\epsilon < 1$.
> >Is there a theoretical guarantee that the splits created using the method presented in Section 3.1 will be biased as wished?
>
> We will provide a rough sketch of this theoretical guarantee. The first quality is satisfied by construction since our splits only depend on $U_{i,\ell}$ (which is a function of $G_{i,\ell}$). The second quality holds true per our answer to your first question. Finally, our final choice of $p(U_{i,\ell}|S_{i,\ell})$ (described in the answer to your first point) allow us to control for the amount of bias we want by adjusting the $\epsilon$ hyperparameter such that if $\epsilon < 1$ we are going to get some bias in our splits. We hope this helps clarify the theory and will include more formal proofs in the final version of the paper if accepted.
> > are there any evaluation methods/metrics which have considered images in thin support regions?
>
> That is an interesting question! We are not aware of any such metrics, however the general idea of adapting metrics from the image generation domain to the graph generation domain has been previously explored. For example, Preuer et al. [1] introduced the FCD metric that draws motivation from the FID metric introduced by Heusel et al. [2] in the image generation domain. However, the FCD metric is designed to assess the quality of the molecular data generated by a generative model and isn't a general purpose metric for assessing the quality of all graph structures nor is it designed to evaluate generalizability on thin support. If there exists a metric for assessing the generalization qualities of an image generative model, it could (in theory) be adapted to the graph generation domain. However, we note that image generation is much more developed than graph generation and some metrics may not be easy (or even possible) to adapt.
> > It would be useful to demonstrate in the experiments by showing some examples of new molecules generated
>
> [Here](https://anonymous.4open.science/r/Additions-850B/_UAI_2024__new_addition.pdf) we attempted to visualize the top four highest weighted novel molecules generated by both GDSS and DiGress from two training scenarios created using VV, and add plots to show where they lie with respect to the property distribution. We also show that among the top 100 highest weighted generated molecules, several were found to exactly match those in the held out data, which suggests that these molecules are viable novel molecules. While we hope this helps contextualize our work, assessing whether a generated molecule has potential for medicine or material design is beyond the scope of this paper and the authors' expertise.
>
> [1] K. Preuer, P. Renz, T. Unterthiner, S. Hochreiter, and G. Klambauer. Fréchet chemnet distance: A metric for generative models for molecules in drug discovery. Journal of chemical information and modeling, 2018
>
> [2] M. Heusel, H. Ramsauer, T. Unterthiner, B. Nessler, and B. Hochreiter. GANs trained by a two time-scale update rule converge to a local Nash equilibrium. NeurIPS 2017

---

### Meta-Review · Area_Chair_8pro · 2024-04-17

This paper proposes a new evaluation method called Vertical Validation (VV) for assessing the ability of graph generative models to generate graphs in thin support regions. The idea is to create biased train-test splits where some parts of the data are systematically held out based on a chosen graph property, and use a reweighting strategy to account for the bias in evaluation. However, the clarity (uKVR) and the novelty of the idea is questioned (TMBC). Fortunately, the authors have addressed the main issues proposed by the reviewers and the overall assessment is positive.